# ADAP1 promotes latent HIV-1 reactivation by selectively tuning KRAS–ERK–AP-1 T cell signaling-transcriptional axis

Nora-Guadalupe P. Ramirez [1], Jeon Lee [2], Yue Zheng[3], Lianbo Li [4,5], Bryce Dennis[4,5], Didi Chen[1], Ashwini Challa[1], Vicente Planelles[3], Kenneth D. Westover [4,5], Neal M. Alto [1] & Iván D'Orso [1✉]

Immune stimulation fuels cell signaling-transcriptional programs inducing biological responses to eliminate virus-infected cells. Yet, retroviruses that integrate into host cell chromatin, such as HIV-1, co-opt these programs to switch between latent and reactivated states; however, the regulatory mechanisms are still unfolding. Here, we implemented a functional screen leveraging HIV-1's dependence on CD4$^+$ T cell signaling-transcriptional programs and discovered ADAP1 is an undescribed modulator of HIV-1 proviral fate. Specifically, we report ADAP1 (ArfGAP with dual PH domain-containing protein 1), a previously thought neuronal-restricted factor, is an amplifier of select T cell signaling programs. Using complementary biochemical and cellular assays, we demonstrate ADAP1 inducibly interacts with the immune signalosome to directly stimulate KRAS GTPase activity thereby augmenting T cell signaling through targeted activation of the ERK–AP-1 axis. Single cell transcriptomics analysis revealed loss of ADAP1 function blunts gene programs upon T cell stimulation consequently dampening latent HIV-1 reactivation. Our combined experimental approach defines ADAP1 as an unexpected tuner of T cell programs facilitating HIV-1 latency escape.

[1] Department of Microbiology, The University of Texas Southwestern Medical Center, Dallas, TX 75390, USA. [2] Lyda Hill Department of Bioinformatics, The University of Texas Southwestern Medical Center, Dallas, TX 75390, USA. [3] Division of Microbiology and Immunology, Department of Pathology, University of Utah School of Medicine, Salt Lake City, UT 84112, USA. [4] Department of Biochemistry, The University of Texas Southwestern Medical Center, Dallas, TX 75390, USA. [5] Department of Radiation Oncology, The University of Texas Southwestern Medical Center, Dallas, TX 75390, USA. ✉email: Ivan.Dorso@utsouthwestern.edu

Immune stimulation activates precisely orchestrated signaling-transcriptional programs (effector programs) facilitating immune cell transitions from homeostatic to effector states[1,2]. Dysregulation of effector programs can result in poor pathogen clearance (e.g., due to nonresponsive immune cells), collateral damage to host tissues (e.g., due to chronic inflammation), or oncogenic progression (e.g., due to uncontrolled clonal expansion). Counterintuitively, while effector programs are typically mounted to eliminate pathogenic infections, retroviruses that permanently integrate into host cell chromatin, such as HIV-1, capitalize on these programs for gene expression and replication.

Though current anti-retroviral therapy suppresses actively replicating HIV-1, latent proviruses persist indefinitely in resting memory CD4$^+$ T cells (T$_M$)[3–5]. Critically barring eradication efforts, infection can be re-established upon therapy cessation due to the expansion of T cells harboring latent yet replication-competent proviruses[6], motivating the need for understanding effector programs licensing HIV-1 persistence. However, it remains poorly understood which inducible host cell factors regulate these effector programs. This gap in knowledge offers an unparalleled opportunity to discover factors and regulatory mechanisms that can illuminate alternative therapeutic strategies for eliminating latently infected cells.

Our studies stem from the premise that during T cell stimulation, several factors including sequence-specific transcription factors (TFs) recognizing cellular and viral genes, are induced, stabilized, or re-localized to subcellular compartments that drive signaling pathways fostering latent viral reactivation (Fig. 1a)[7]. Given HIV-1 proviral fate is contingent on T cell transcriptional activation, we reasoned that exploiting HIV-1's dependence on T cell status, by systematically over-expressing human factors in a cell-based model of HIV-1 latency (Fig. 1b), will allow identification of inducible factors promoting T cell transcriptional programs and consequently latent HIV-1 reactivation. The expectation was that various degrees of activation will be uncovered depending on the factor's mode of action on the cell signaling-transcriptional cascade (e.g., initiating signaling events in the plasma membrane, cytoplasm and/or nucleus).

Here, we present the discovery and detailed studies of ADAP1 as a previously unappreciated tuner of T cell signaling-transcriptional programs facilitating HIV-1 to escape latency. We find ADAP1 is an inducible factor interacting with the immune signalosome to selectively amplify the ERK–AP-1 axis. Interestingly, ADAP1-mediated ERK–AP-1 activation is dependent on direct stimulation of the KRAS GTPase, a function not previously reported. Consistently, loss of ADAP1 function coupled with single cell transcriptomic analysis support a model whereby ADAP1 tunes T cell gene programs ultimately influencing HIV-1 transcriptional activation potential, thus implicating ADAP1 as an undescribed T cell signaling-transcriptional tuner that modulates HIV-1 proviral fate.

## Results
**Gain-of-function screen reveals ADAP1 as a latent HIV-1 activating factor with undescribed T cell signaling functions.** To identify HIV-1 activating host cell factors (Fig. 1b), we developed an unbiased gain-of-function screening strategy (Fig. 1c), in which human cDNAs were delivered into a Jurkat T cell line bearing an inducible HIV-1 luciferase reporter (Jkt-HIVLuc) as a proxy for virus reactivation from latency (Supplementary Fig. 1a, b). Candidate factors were then validated and characterized in primary CD4$^+$ T cells to provide physiologic relevance.

Briefly, a human cDNA library ($n = 17,384$ cDNAs) was divided into 183 pools of 96 cDNAs per pool, which were used for generating lentiviral pools to transduce Jkt-HIVLuc cells (Fig. 1c, Phase 1). Latent HIV-1 reactivation was evaluated using luciferase assays in two independent screens. Inherently, the 1 in 96 cDNA dilution per pool set a high-stringency threshold of discovery, as theoretically only ~1% of cells will express an individual activating factor. Given the high-stringency nature of the screen, we selected a relaxed threshold (fold change > 1) for initially calling positive pools. Using this criteria, 69 positive pools were identified in the two parallel screens (FDR < 2.5%, multiple unpaired $t$-test) (Fig. 1d and Supplementary Fig. 1c) as potentially containing a cDNA encoding a latent HIV-1 activating factor. Among these 69 pools, 15 were prioritized based on a cross-referencing selection criterion between screens (Supplementary Fig. 1d) and further resolved into smaller pools to identify the individual latent HIV-1 activating factor within each pool (Fig. 1c, Phases 2 and 3 and Supplementary Fig. 1e, f).

Collectively, decomposition of the 15 pools revealed 7 candidate factors inducing latent HIV-1 > 2-fold (TNFα, CD80, CD86, KRAS4b ("KRAS"), CD40, TIRAP, ADAP1) in addition to LTA, which was below the significance threshold (Fig. 1e). Among these hits were well known molecules that initiate upstream signaling pathways such as ligands (TNFα, CD80, CD86), receptors (CD40), and others that potentiate signaling originating at the plasma membrane such as transducer (KRAS) and adapter (TIRAP) proteins[8–12]. Further, protein association analysis using STRING[13] reinforced 7 candidates (TNFα, CD80, CD86, KRAS, CD40, TIRAP, LTA) were enriched in immune functions, associated with multiple immune cell regulators (Fig. 1f), and implicated in the context of HIV-1 infection[14–17]. However, strikingly, ADAP1 (ArfGAP with dual PH domain-containing protein 1, also known as Centaurin-Alpha-1), originally identified as a neuronal-restricted factor[18,19], was excluded from the protein association analysis (Fig. 1f). Intriguingly, publicly available genome-wide association studies found *ADAP1* single nucleotide polymorphisms (SNPs) were related to altered lymphocyte counts[20], offering clinical relevance and warranting the study of ADAP1 as a previously overlooked regulator of the immune system, and consequently exploited by HIV-1.

To define the mechanism by which ADAP1 contributes to T cell signaling for latent HIV-1 reactivation, we first performed domain mapping analysis. ADAP1 contains three domains, an Arf GTPase activating protein (GAP) domain that tetrahedrally coordinates metal (Zn) and two membrane-binding pleckstrin homology domains (PH1 and PH2) (Fig. 1g)[21]. By ectopically expressing previously described GAP (C24A, GAP Mut) and membrane-binding (R149C, PH1 Mut) deficient mutants[22] in Jkt-HIVLuc cells, which are *ADAP1* deficient (Fig. 1g), we found both mutants had statistically significant decreases in luciferase activity when compared to wild-type (WT) ADAP1 (~1.7-fold, and ~1.4-fold respectively, $p < 0.01$, paired $t$-test) (Fig. 1h), signifying ADAP1 requires both GAP domain and plasma membrane-binding to promote latent HIV-1 reactivation. Unexpectedly, the GAP mutant expressed at lower levels compared to WT ADAP1 (Fig. 1g), potentially attributed to a dual requirement for GAP activity in protein folding through metal cofactor coordination[21] and protein stability, consistent with the destabilization of other enzymes upon catalytic core inactivation[23]. Together, while both GAP and PH1 domains appear to be required for ADAP1-mediated latent HIV-1 reactivation, the virtual loss of GAP activity cannot be fully uncoupled from the expression loss.

Surprisingly, ADAP2, which is ~55% identical to ADAP1 and is induced by interferon to block RNA viral (e.g., DENV, VSV) infections[24], does not reactivate latent HIV-1 (Supplementary Fig. 1g), thus revealing intriguing functional specificity among ADAP family members.

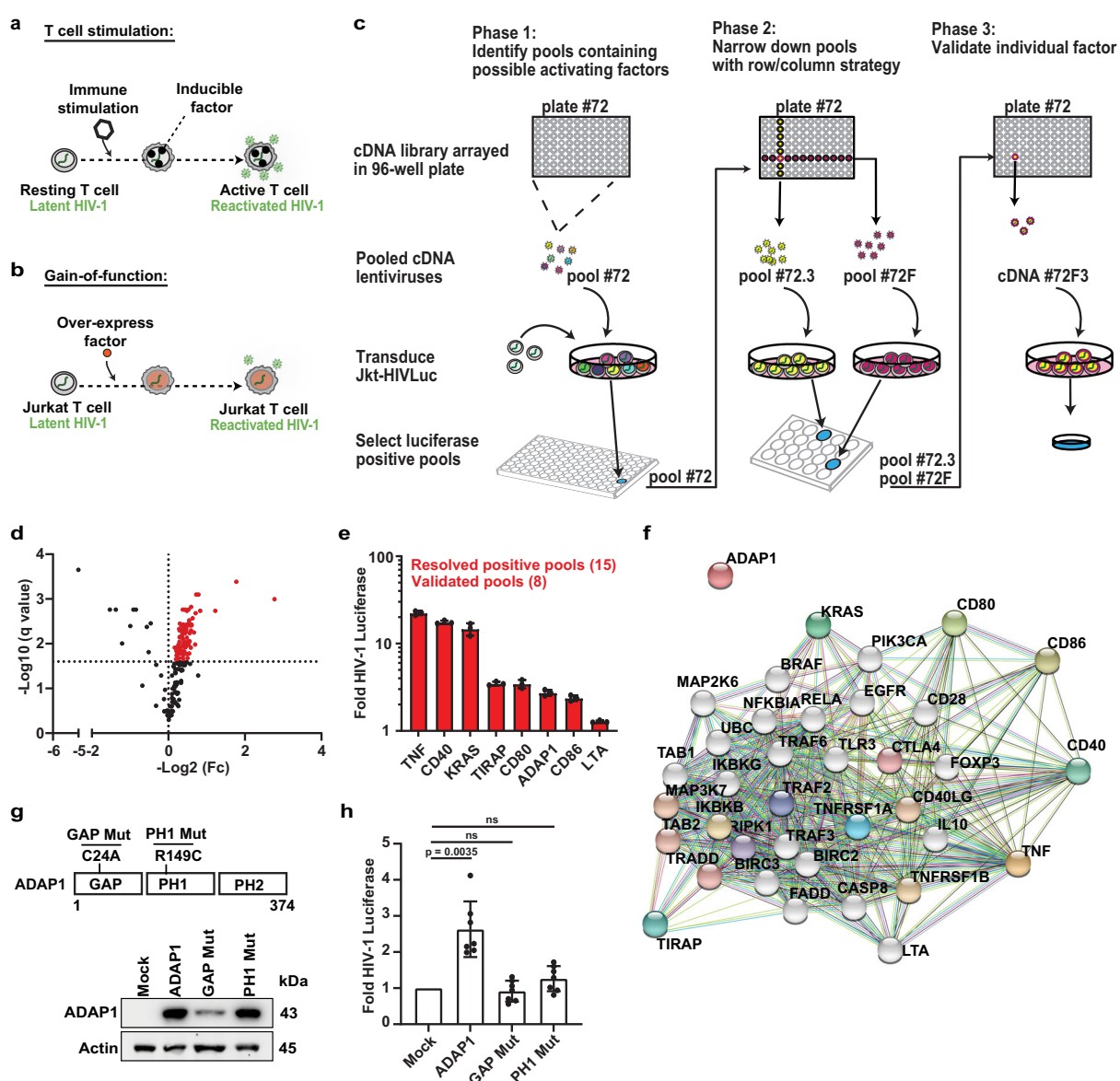

**Fig. 1 Gain-of-function screen reveals ADAP1 as a latent HIV-1 activating factor with undescribed T cell signaling functions. a** Schematic of immune stimulation-mediated CD4$^+$ T cell gene expression coupled to HIV-1 gene expression motivating the screen in **b**. **b** Gain-of-function screen approach to discover undescribed T cell signaling-transcriptional regulators relevant for latent HIV-1 reactivation and T cell programs. **c** Schematic of 3-phase screen approach. Pool 72 is highlighted as example. See "Methods" for screen details. **d** Volcano plot summarizing results of phase 1 of gain-of-function screen (repeat 1). Each dot is a pool sample (representing 96 cDNAs/pool) mean luciferase activity ($n = 3$) and statistical significance was determined using a multiple unpaired $t$-test. Dots in red denote positive pools above the chosen cutoff of Log$_2$Fc > 0 with an FDR < 2.5% and cross-referenced with the results of screen repeat 2 (Supplementary Fig. 1c) to determine which pools to resolve (see Supplementary Fig. 1d details). **e** Summary of candidate HIV-1 activating factors first identified in the gain-of-function screen and then validated individually. Data represents mean ± s.d. of three independent experiments ($n = 3$). **f** Candidates from **e** were entered into STRING (Search Tool for the Retrieval of Interacting Genes/Proteins), a search database that identifies known and predicted protein–protein functional and direct associations and mined for predicted functional interactions based on available databases (represented by different colored edges). Colored nodes are inputted list from **e** and first shell of predicted interactors, while white nodes are secondary interactors. Figure displayed contains 30 additional nodes with the inputted 7 nodes. **g** Domain mapping analysis. Top: schematic of ADAP1 domains and point mutations used for further analysis. Bottom: western blot of Jkt-HIVLuc cells transduced with lentiviruses expressing empty vector (Mock), wild-type (WT) ADAP1, or point mutants (GAP Mut, PH1 Mut), and probed with the indicated antibodies. **h** Fold luciferase activity of Jkt-HIVLuc cells transduced in **g**. Data represents mean ± s.d. of seven independent experiments ($n = 3$) (one-way ANOVA followed by Dunnett's test for multiple comparisons to mock). ns not significant. Source data are provided as a Source Data file.

**ADAP1 is expressed in primary human CD4$^+$ T cells.** Since ADAP1 expression and function in immune cells are unknown, and because the above data was generated in an immortalized model of latency under overexpression conditions, we expanded our studies to incorporate physiologically relevant systems. Specifically, we focused on primary CD4$^+$ T cells which undergo

drastic transcriptional changes during cell state transitions (from naïve (T$_N$) to effector (T$_E$), from T$_E$ to resting memory (T$_M$), and from T$_M$ to stimulated state (stimulated T$_M$)) (Fig. 2a). Additionally, both T$_N$ and T$_M$ are the most abundant cell types comprising the latent HIV-1 reservoir in infected individuals[25]. To characterize *ADAP1* expression during T cell state transitions,

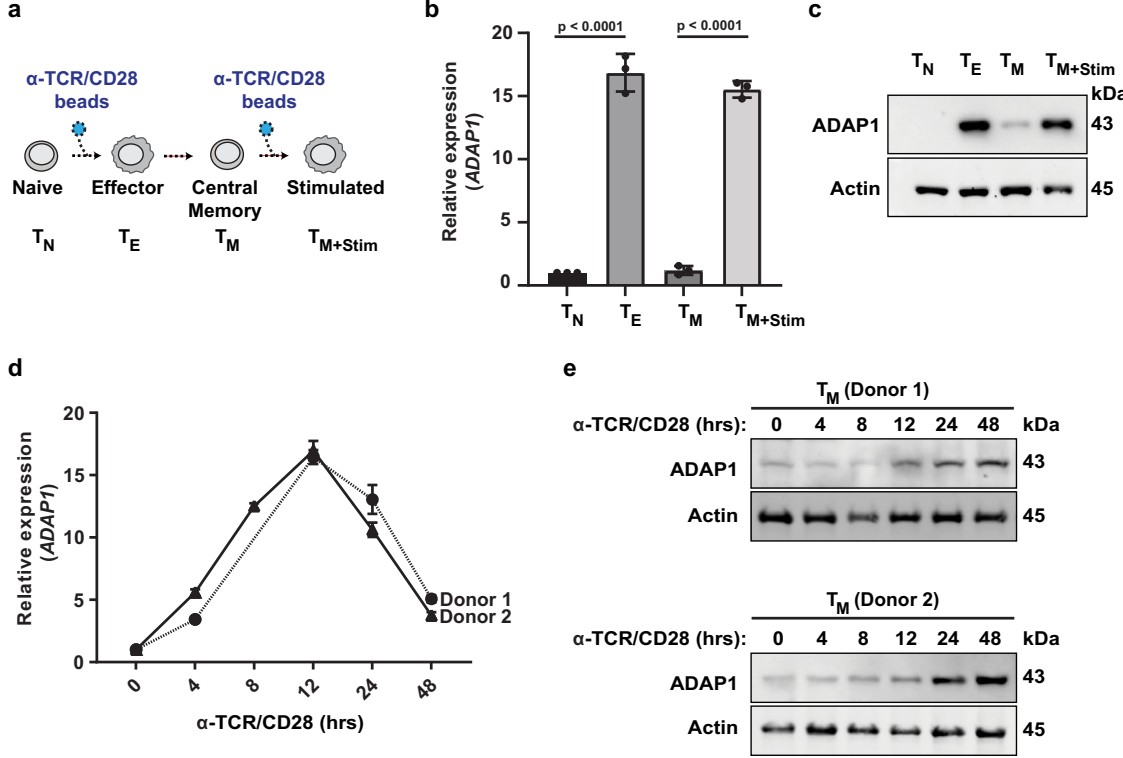

**Fig. 2 ADAP1 is expressed in primary human CD4$^+$ T cells. a** Schematic of primary CD4$^+$ T cell states generated and analyzed in the subsequent panels. **b** Relative mean ± s.d. ($n = 3$) mRNA expression of *ADAP1* in one representative donor across differing T cell states (repeated in two other donors with similar results). $T_N$ were sampled on the day of isolation, $T_M$ were sampled 18–21 days post donor isolation depending on when they reached quiescence, $T_E$ and $T_{M+Stim}$ were samples stimulated for 24 h. (one-way ANOVA followed by Dunnett's test for multiple comparisons to mock). **c** Representative western blot analysis of ADAP1 expression in primary CD4$^+$ T cells across differing states. $T_N$ were sampled on the day of isolation, $T_M$ were sampled 18–21 days post isolation depending on when they reached quiescence, $T_E$ and $T_{M+Stim}$ were samples stimulated for 24 h (repeated in two other donors with similar results). **d** Relative mean ± s.e.m. ($n = 3$) mRNA expression of *ADAP1* in two representative donors across differing times post-stimulation. **e** Representative western blot analysis of ADAP1 expression in two representative donors across differing times post-stimulation. Source data are provided as a Source Data file.

we first isolated human peripheral $T_N$ from healthy donors and generated the other T cell states which were validated by staining cells with anti-CD4 (for purity assessment), anti-CD45RA (naïve marker), anti-CD45RO (memory marker), and Ki67 (proliferation marker) (Supplementary Fig. 2a, b). We then measured *ADAP1* RNA and protein expression levels in each state using RT-qPCR and western blot assays, respectively. Notably, we found antigenic (T cell receptor (TCR)/CD28) stimulation of $T_N$ and $T_M$ for 24 h (Fig. 2a) induced *ADAP1* RNA (~15-fold) and protein (~12-fold) relative to unstimulated cells (Fig. 2b, c), suggesting ADAP1 is a T cell inducible protein potentially modulating key functions during T cell activation that are co-opted by latent HIV-1 to promote its reactivation. To define the kinetics of ADAP1 induction we performed a time-course stimulation of $T_M$ from two donors. *ADAP1* RNA expression gradually increased at 4 and 8 h until reaching maximum induction at 12 h post-stimulation and then gradually decreased at 24 and 48 h post-stimulation (Fig. 2d). Consistently, ADAP1 protein levels in the two donors showed an increase at 12 h (~1.7 to 3-fold) reaching a maximum at 24 and 48 h (~2.4 to 5.3-fold) post-stimulation (Fig. 2e).

**Loss of *ADAP1* in primary CD4$^+$ T cells hinders latent HIV-1 reactivation ex vivo.** The discovery that ADAP1 ectopic expression induced HIV-1 gene expression from Jkt-HIVLuc cells, prompted us to test whether loss of *ADAP1* in primary T cells compromised latent HIV-1 reactivation. For this, we

used a primary model of latency in where latently infected $T_M$ were CRISPR-Cas9 genetically depleted of *ADAP1* (HIV-ADAP1$^{CRISPR}$), negative control (HIV-Ctrl$^{CRISPR}$), surface protein gene *CXCR4* as additional negative control (HIV-CXCR4$^{CRISPR}$), or *RELA* the p65 subunit of NF-κB as a positive control (HIV-NF-κB$^{CRISPR}$) (Fig. 3a, b). Stimulation with phorbol myristate acetate (PMA) for 24 h across three donors, indicated loss of *ADAP1* reduced (~2-fold) latent HIV-1 reactivation relative to HIV-Ctrl$^{CRISPR}$ and HIV-CXCR4$^{CRISPR}$ negative controls (Fig. 3c), complementing the ectopic expression data in Jkt-HIVLuc cells (Fig. 1h). Remarkably, this phenotype was nearly identical to NF-κB loss (Fig. 3c), an established HIV-1 positive regulator[7], signifying ADAP1's importance to HIV-1 gene expression. Further, unexpectedly, resting HIV-ADAP1$^{CRISPR}$ $T_M$ exhibited slightly, but statistically significant, elevated HIV-1 expression prior to stimulation (~1.3-fold luciferase) relative to HIV-Ctrl$^{CRISPR}$ and HIV-CXCR4$^{CRISPR}$ cells, that was not observed with NF-κB loss (Supplementary Fig. 3).

**ADAP1 localizes to the plasma membrane where it interacts with the immune signalosome in primary CD4$^+$ T cells.** Since immune regulators are typically located in the plasma membrane to impart cell signaling changes and because ADAP1 requires plasma membrane binding for latent HIV-1 reactivation (Fig. 1h), we next examined ADAP1 subcellular localization using immunofluorescence staining and cell fractionation. First,

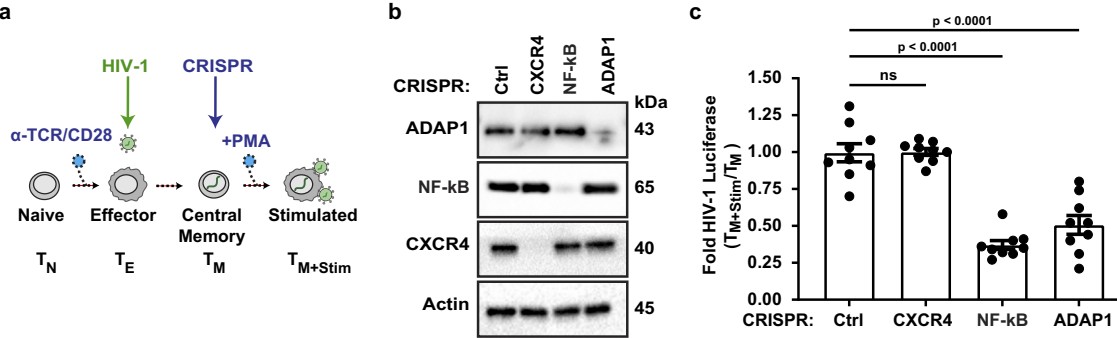

**Fig. 3 Loss of *ADAP1* hinders latent HIV-1 reactivation ex vivo. a** Schematic of generation of ex vivo model of HIV-1 latency and CRISPR-Cas9 engineered cells HIV-Ctrl[CRISPR], HIV-CXCR4[CRISPR], HIV-NF-κB[CRISPR](p65 subunit), and HIV-ADAP1[CRISPR]. **b** Representative western blot analysis of CRISPR-Cas9-mediated knockout efficiency (repeated in two other donors with similar results). **c** Luciferase analysis of HIV-Ctrl[CRISPR], HIV-CXCR4[CRISPR], HIV-NFκB[CRISPR], and HIV-ADAP1[CRISPR] $T_M$ stimulated for 48 h with PMA. Data represents mean ± s.d. fold change of three donors ($n = 3$ each), HIV-Ctrl[CRISPR] normalized to 1 representing max activation. (one-way ANOVA followed by Dunnett's test for multiple comparison to Ctrl[CRISPR]). ns not significant. Source data are provided as a Source Data file.

to be able to simultaneously compare T cell states, we imaged $T_M$ with and without TCR/CD28 stimulation (4 h) in which ADAP1 expression levels are similar (Fig. 2e) and found that ADAP1 is localized at the cell periphery in both states (Fig. 4a). Next, to further determine whether ADAP1 peripheral localization is cytoplasmic and/or membrane-bound and to more precisely evaluate potential ADAP1 subcellular re-localization during T cell stimulation, we fractionated $T_M$ with and without TCR/CD28 stimulation in a time-course (1, 4, and 24 h) using established protocols[26,27]. Cell fractionation was first validated by examining the partitioning of proteins exclusively detected in cytoplasm (GAPDH) or plasma membrane (CD3ε), and a protein known to shuttle between cytoplasm and plasma membrane upon T cell stimulation (PKCθ). Having validated the cell fractionation protocol, we then examined ADAP1's localization and found ADAP1 was largely cytosolic prior to stimulation and became re-localized to the plasma membrane during the time-course TCR/CD28 stimulation (Fig. 4b), consistent with the potential regulation of T cell signaling by ADAP1 plasma membrane binding upon T cell stimulation.

Given the findings that ADAP1's plasma membrane localization increased upon T cell stimulation (1, 4 and 24 h) (Fig. 4b), we predicted ADAP1 would interact with the immune signalosome. To test this idea, ADAP1 was immunoprecipitated (IP) from resting and TCR/CD28-stimulated (1 h) $T_M$ followed by profiling interacting partners using tandem mass spectrometry. Strikingly, in protein lysates from two donors, ADAP1 co-purified with components of the early T cell signalosome (e.g., LCK, PKCθ, ZAP70, PI3K)[28] only upon stimulation (Fig. 4c, d and Supplementary Data 1 and 2). Protein interaction network analysis further highlighted ADAP1 co-factors were enriched in functional clusters including TCR signaling, activation of immune responses and, interestingly, host–HIV-1 interactions and viral life cycle (Fig. 4e).

ADAP1 interaction with PKCθ (Fig. 4c, d) was of notable interest as PKCθ is essential for TCR-mediated T cell activation[29], transiently activating multiple effectors (e.g., RasGRP1[30] and CARMA1[31–33], which drive signaling cascades inducing the TFs AP-1[30] and NF-κB[29,31,34], respectively). Notably, PKCθ-ADAP1 interaction upon T cell stimulation was validated by IP-western blot (Fig. 4f) and is consistent with a direct protein–protein interaction in vitro[35]. Collectively, these results suggest ADAP1 is recruited to the immune signalosome likely where it interacts with early regulators of T cell signaling.

**Loss of *ADAP1* in primary CD4+ T cells dampens transcriptional programs upon stimulation.** Since HIV-1 gene expression is induced by ADAP1 ectopic expression in Jurkat cells (Fig. 1h) and ADAP1 interacts with the immune signalosome in response to stimulation in primary T cells (Fig. 4c, d), we reasoned ADAP1 may amplify select T cell signaling-transcriptional programs. To identify these ADAP1-regulated gene programs, transcriptomes of physiologically relevant T cells engineered through CRISPR-Cas9–mediated *ADAP1* ablation (ADAP1[CRISPR]) and negative control (Ctrl[CRISPR]) in both resting and TCR/CD28-stimulated (4 h) $T_M$ states (Fig. 5a, b and Supplementary Fig. 4) were profiled through single cell (sc)RNA-seq, which enables clustering of seemingly identical cells from a heterogeneous, non-clonal population of resting and activated lymphocytes.

Stimulation of Ctrl[CRISPR] $T_M$ revealed 882 differentially expressed genes (DEGs) above $\log_2Fc > 0.25$ cutoff, of which 626 (71%) were upregulated (referred to as induced genes) and included well known T cell activated genes (e.g., *CD69*, *IL2*) (Fig. 5c and Supplementary Data 3). Clustering of cells harboring similar gene expression patterns revealed the identified induced genes were preferably enriched in one cluster that only appeared after stimulation (activated cluster, Fig. 5d). The loss of *ADAP1* reduced the expression of 397 out of the 626 (63.4%) induced genes originally identified in stimulated Ctrl[CRISPR] $T_M$ as revealed by linear regression of all genes at the bulk level (Fig. 5e) and by violin plots of individual genes at the single cell (Fig. 5f). Interestingly, the remainder of the T cell activated genes (229 out of 626 (36.6%)) that were not induced above the $\log_2Fc > 0.25$ cutoff in stimulated ADAP1[CRISPR] $T_M$ (Fig. 5c), included secreted growth and differentiation cytokines (e.g., *IL2*, *CSF2*). In contrast, these genes were highly induced in stimulated Ctrl[CRISPR] $T_M$ ($\log_2Fc > 2.7$, 2.4 respectively) (Supplementary Data 3), suggesting *ADAP1* loss triggers defects in normal T cell growth and activation, potentially explaining the association of *ADAP1* SNPs to altered T cell counts[20]. Expectedly, gene ontology (GO) analysis revealed the induced genes were functionally enriched in biological processes such as T cell activation, regulation of cytokine production, myeloid cell differentiation, and leukocyte activation involved immune response (Fig. 5g), signifying ADAP1 contributes to the normal activation of T cell programs. Importantly, the induced genes contained *cis*-elements enriching for canonical TCR-induced TFs (e.g., AP-1, NF-κB) (Fig. 5h), which are both important for HIV-1 and T cell transcription activation.

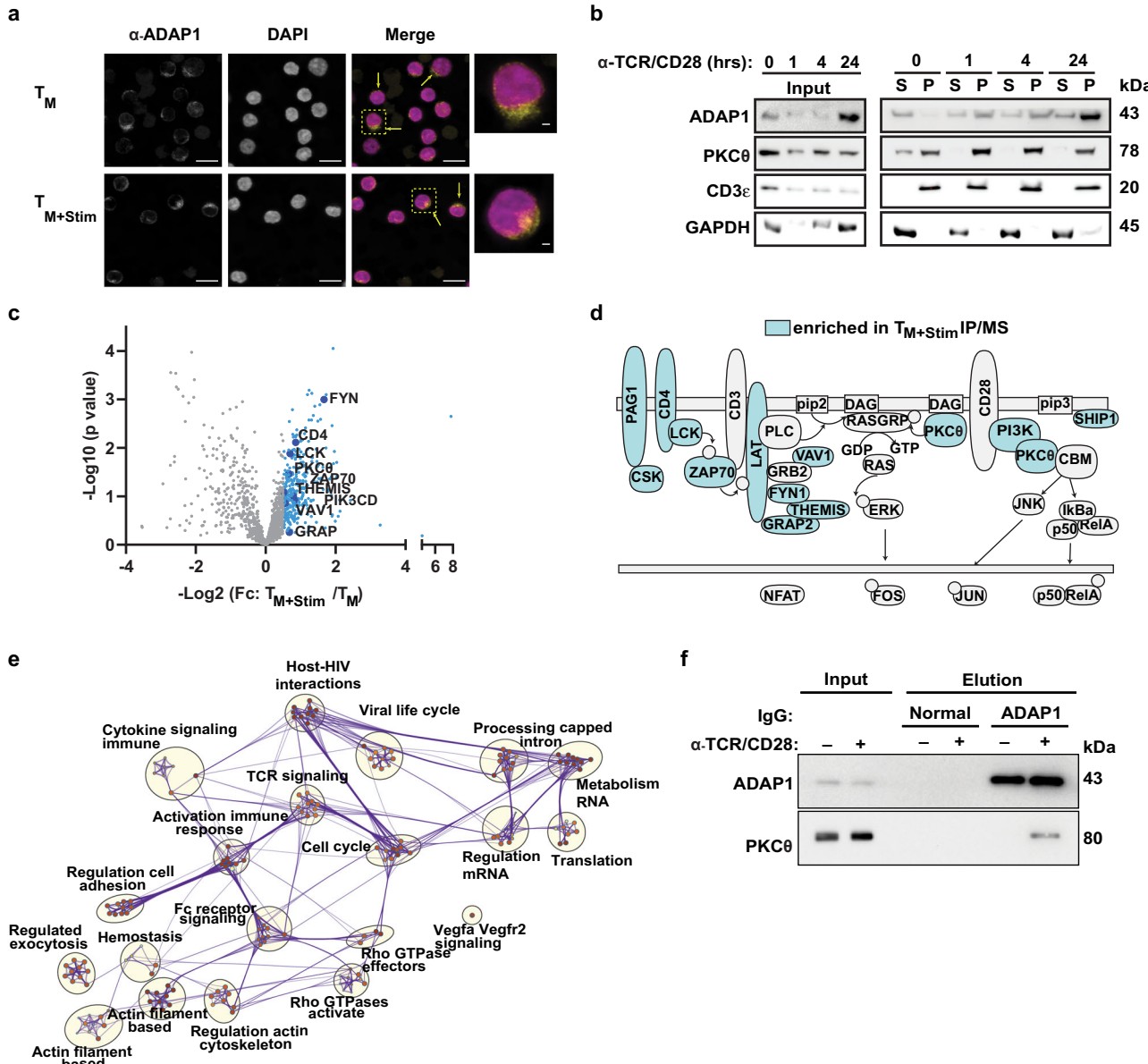

**Fig. 4 ADAP1 interacts with the immune signalosome. a** Representative immunofluorescence probing ADAP1 localization in unstimulated and stimulated (4 h TCR/CD28) $T_M$. Repeated in 1 other donors with similar results. Scale bar is 10 μm for multicell images and 1 μm for single cell images. **b** Representative donor ($n = 2$) western blot analysis of cell fractionation assay. Resting (0 h) or stimulated $T_M$ (1 h, 4 h, 24 h) were fractioned into "S" supernatant (cytosolic) or "P" pellet (membrane) fraction. **c** Resting $T_M$ and stimulated (1 h) $T_M$ of one donor were immunoprecipitated in triplicate with ADAP1 antibody and submitted for tandem mass spectrometry analysis. Volcano plot represents average $Log_2Fc$ of factor enrichment in stimulated $T_M$ over resting $T_M$. Samples with $Log_2Fc > 0.5$ are highlighted in blue. A sample of immune signalosome proteins are highlighted. See Supplementary Data 1 for complete list of interacting partners, semi-quantitative abundance and statistical significance. **d** Visual representation of second donor mass spectrometry results of $T_M$ stimulated for 1 h and immunoprecipitated with IgG or ADAP1 antibodies. See Supplementary Data 2 for complete list of interacting partners, semi-quantitative abundance. Interacting proteins that form part of the immune signalosome or T cell membrane are highlighted in blue. **e** Protein–protein interacting network analysis clustered by gene ontology. Nodes are color-coded and size-coded based on $p$ value. Edges (purple lines) represent network connections. Gene ontology category derived from Metascape analysis are written next to clusters. **f** Representative western blot analysis of resting or stimulated $T_M$ immunoprecipitated with ADAP1 or IgG antibodies ($n = 2$ donors). Samples were used for tandem mass spectrometry analysis (**d**, **e**). Elutions were additionally probed with PKCθ to validate mass spectrometry results. Source data are provided as a Source Data file.

**ADAP1 promotes KRAS-dependent ERK–AP-1 axis activation.** Since TCR-induced and HIV-1 relevant TFs were enriched in transcriptome data, we next sought to define ADAP1-regulated TFs. For this, we ectopically expressed ADAP1 in Jurkat cells, which are *ADAP1* deficient (Fig. 1g), alongside luciferase reporters containing *cis*-elements for the three canonical TCR and HIV-1 responsive TFs (AP-1, NFAT, and NF-κB) in the absence of any stimulation. Notably, ADAP1 ectopic expression induced

AP-1, but not NF-κB nor NFAT, reporter activity (~1.5-fold, $p < 0.001$, two-way ANOVA with multiple comparisons) (Fig. 6a), consistent with increased (~2-fold) AP-1 subunits (Fos and Jun) levels in ADAP1-expressing Jurkat cells (Fig. 6b). While the increase in both AP-1 reporter activity and AP-1 protein levels are small, this could be explained by the intrinsic AP-1 signaling in Jurkat, consistent with high reporter levels in the absence of ADAP1 ectopic expression (Fig. 6a). Additionally, residual levels

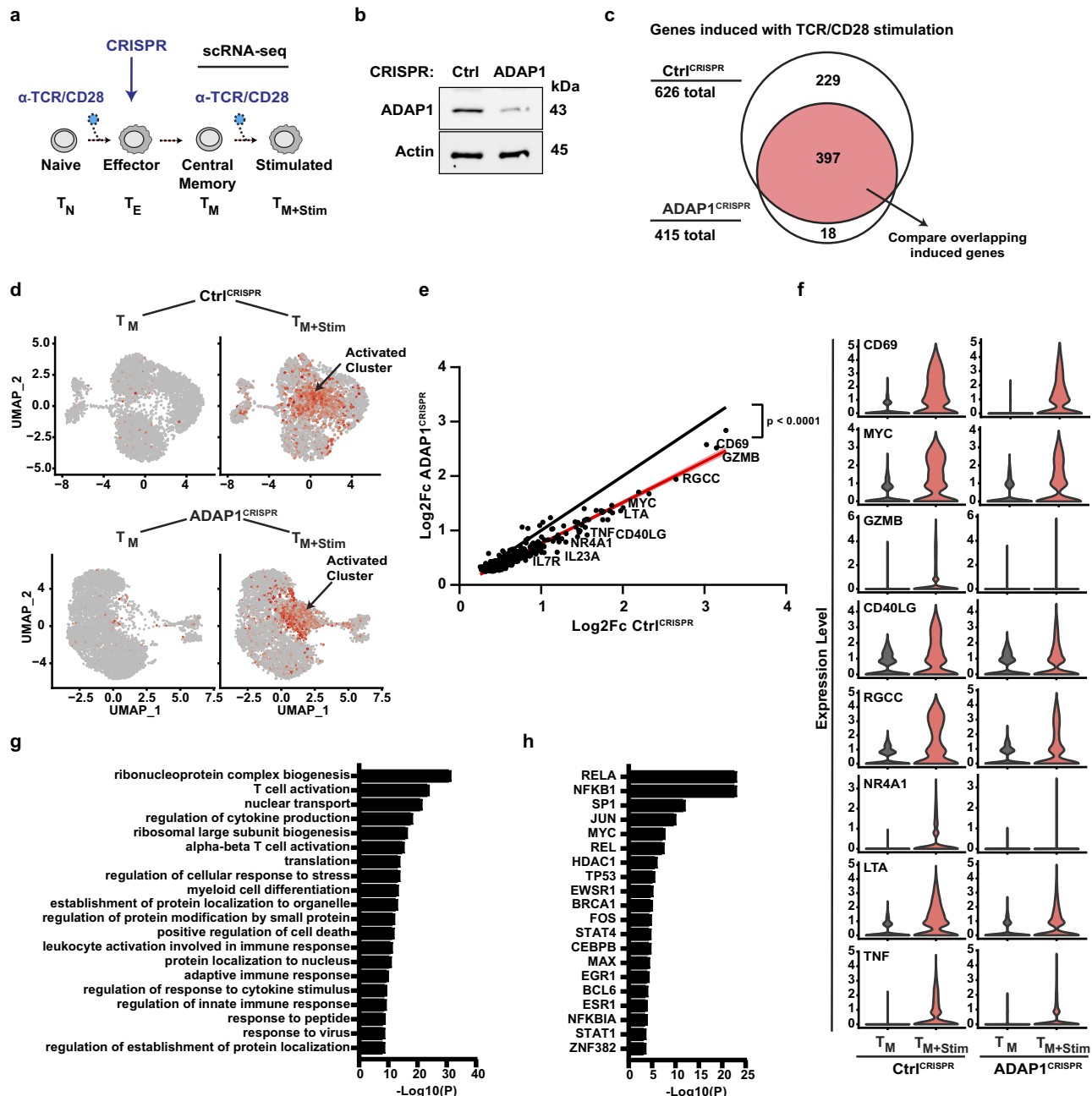

**Fig. 5 Loss of *ADAP1* in primary human CD4$^+$ T cells dampens transcriptional gene programs upon stimulation. a** Schematic of the generation of ADAP1$^{CRISPR}$ and Ctrl$^{CRISPR}$ T$_M$ subjected to scRNAseq analysis. **b** Representative western blot analysis of primary T$_M$ (ADAP1$^{CRISPR}$ and Ctrl$^{CRISPR}$) engineered through CRISPR-Cas9 ($n = 4$ donors). **c** scRNAseq analysis of resting and stimulated (4 h) ADAP1$^{CRISPR}$ and Ctrl$^{CRISPR}$ T$_M$. Differentially expressed genes (DEG) were identified by comparing resting and stimulated T$_M$ within each sample set. Results of induced genes are summarized in Venn diagram. The overlapping induced genes (red) are those upregulated in both ADAP1$^{CRISPR}$ and Ctrl$^{CRISPR}$ samples sets albeit at different levels (see text for details). **d** Induced genes were used to compare level of fold changes between ADAP1$^{CRISPR}$ and Ctrl$^{CRISPR}$ samples. Select gene markers (highlighted in **e**, **f**), were used to visualize the cells (red dots) expressing induced genes which mapped to a cluster that appeared during stimulation and are referred to as activated cluster. **e** Graph of induced genes ($n = 397$) found in both ADAP1$^{CRISPR}$ and Ctrl$^{CRISPR}$ sample sets comparing level of Log$_2$fc in respective samples. Using linear regression and *F*-test, the slope of sample data sets (red line, $r^2 = 0.9285$, $Y = 0.7559X + 0.001903$, $F(DFn, DFd) = 535.2 (1, 395)$, 95% confidence in light shaded red) was compared to the null hypothesis (black line, $r^2 = 1.000$, $Y = 1.000X$) of no difference in gene induction between ADAP1$^{CRISPR}$ and Ctrl$^{CRISPR}$ samples. Select induced genes were highlighted. **f** Violin plots displaying mean expression level of induced genes at the single cell level (resting T$_M$ are in gray and stimulated T$_M$ are in red). Genes are also highlighted in **e**. **g** Gene ontology and **h** predicted TFs regulating the differentially expressed, induced genes. Analysis performed using ToppGene Suites[53]. Source data are provided as a Source Data file.

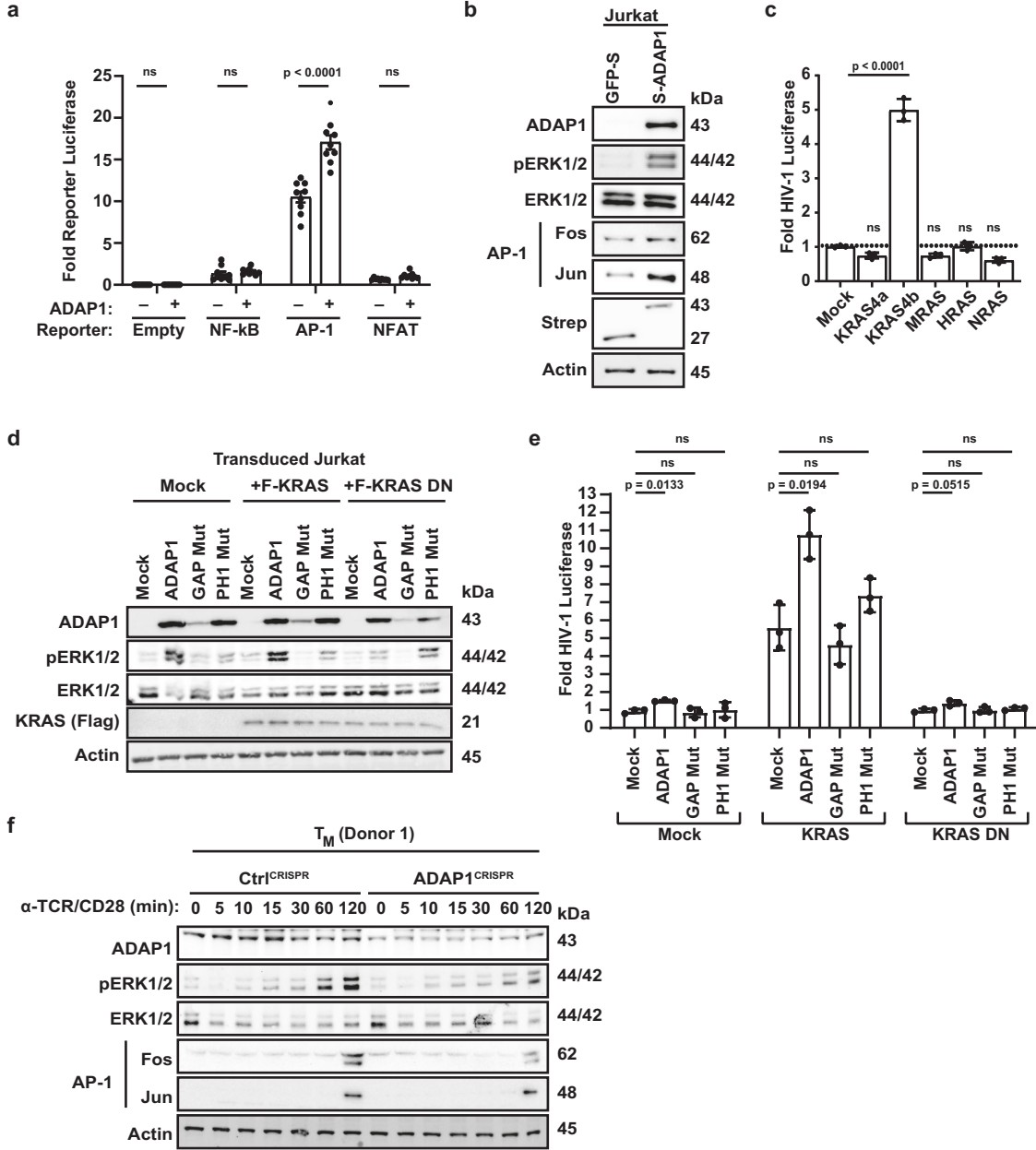

**Fig. 6 ADAP1 promotes ERK–AP-1 activation in a KRAS-dependent manner. a** Jurkat cell lines expressing Strep-tagged ADAP1 or GFP were electroporated with TF-responsive element driving Firefly luciferase reporters. Firefly luciferase activity was measured 48 h post electroporation normalized to a CMV-Renilla control. Data represents mean ± s.d. of three independent electroporation experiments in triplicate (two-way ANOVA followed by Sidak's test for multiple comparison between ADAP1− and ADAP1+ groups). ns not significant. **b** Western blot analysis of Jurkat cell lines expressing Strep-tagged ADAP1 or GFP in the absence of stimulation. Blots are representative of three independent experiments. **c** Fold luciferase activity comparing RAS family members in transduced Jkt-HIVLuc model. Data represents mean ± s.d. fold luciferase activity ($n = 3$). (one-way ANOVA followed by Dunnett's test for multiple comparisons to mock). ns not significant. **d** Western blot analysis of Jurkat cells co-transduced with ADAP1 (WT, GAP Mut, or PH1 Mut) and limiting amounts of Flag-tagged KRAS (WT or dominant negative S17N). Blots are representative of three independent experiments. **e** Fold luciferase activity comparing co-expression of ADAP1 (WT, GAP Mut or PH1 Mut) with low level (4:1 lentiviral dilution) KRAS (WT or DN) in transduced Jkt-HIVLuc cells. Data represents mean ± s.d. fold luciferase activity ($n = 3$) (two-way ANOVA followed by Dunnett's test for multiple comparison to mock). ns not significant. **f** Western blot analysis of primary ADAP1[CRISPR] and Ctrl[CRISPR] $T_M$ stimulated with anti-TCR/anti-CD28 beads in a time-course-dependent manner from 0 to 120 min. Blots shown are representative examples from one of four donors. Source data are provided as a Source Data file.

of Fos (AP-1) were observed in unstimulated ADAP1[CRISPR] $T_M$ (Supplementary Fig. 5a, b), perhaps consistent with slightly elevated basal HIV-1 expression observed previously (Supplementary Fig. 3).

The finding that ADAP1 can activate a nuclear TF (AP-1) along with our previous results indicating ADAP1 is an immune

signalosome component (Fig. 4c, d), suggested ADAP1 modulates one or more proteins involved in the plasma membrane-to-nuclear AP-1 signaling cascade. AP-1 activity is induced through MAPK signaling driven by members of the RAS GTPase family and ERK kinase[28]. Interestingly, the RAS member KRAS was found to be a latent HIV-1 activating factor in our screen

(Fig. 1e), which prompted us to test for specificity of activation by other RAS family members (KRAS4b "KRAS", KRAS4a, HRAS, NRAS, MRAS). Strikingly, only KRAS stimulated HIV-1 gene expression from Jkt-HIVLuc cells (Fig. 6c) in agreement with being the only RAS member found in the screen (Fig. 1e). Taken together, given ADAP1 activates AP-1 and because AP-1 is induced via KRAS–ERK, we hypothesized ADAP1 regulates the KRAS–ERK–AP-1 axis.

Supporting an ADAP1–KRAS–ERK–AP-1 model, ADAP1-expressing Jurkat cells had increased ERK site-specific (Thr202/Tyr204) phosphorylation (Fig. 6b). To determine if the ADAP1-mediated ERK phosphorylation was KRAS-dependent, we co-expressed ADAP1 with limiting amounts of KRAS (to minimize KRAS-independent ERK phosphorylation) or with a dominant negative KRAS (S17N) as negative control[36,37], and observed ERK phosphorylation was only stimulated by WT KRAS (Fig. 6d). Additionally, ADAP1 non-functional mutants (Fig. 1g, h) dampened ADAP1-mediated ERK phosphorylation (Fig. 6d). Noteworthy, the functional interplay between ADAP1 and KRAS was further illustrated in the Jkt-HIVLuc cell model. While individual expression of ADAP1 or KRAS activated latent HIV-1 (~1.5-fold and ~6.3-fold, respectively), co-expression of ADAP1 with limiting amounts of KRAS (to avoid excessive ADAP1-independent activation) showed synergistic HIV-1 gene expression (~10.8-fold) of which was lost with non-functional ADAP1 or KRAS mutants (Fig. 6e).

Consistent with increased ERK phosphorylation (~10-fold) upon ADAP1 ectopic expression in Jurkat cells, ADAP1 loss in $T_M$ had reduced ERK phosphorylation (~1.5 to 8-fold) at 30–120 min post-stimulation across four donors thereby indicating loss of sustained signaling (Fig. 6f and Supplementary Fig. 6a–c, e–h). Consequently, Fos (AP-1) induction was also reduced (~2.5 to greater than 10-fold) in response to stimulation across four donors (Fig. 6f and Supplementary Fig. 6a–c, I, j), further capturing ADAP1's effects during activation of primary T cells.

Additionally, since ADAP1 and PKCθ interacted in stimulated $T_M$ (Fig. 4f), and because PKCθ activates the ERK pathway[30], we next tested whether impaired ERK phosphorylation upon ADAP1 loss was detected with direct PKCθ stimulation. For this, we treated $Ctrl^{CRISPR}$ and $ADAP1^{CRISPR}$ $T_M$ with PMA, a PKC agonist, and observed reduced ERK phosphorylation (up to 2-fold) between 5–30 min post-stimulation (Supplementary Fig. 6d), mirroring the result of TCR/CD28-stimulated cells, thus indicating $ADAP1$ loss dampened PKCθ-ERK signaling.

## ADAP1 interacts with and stimulates KRAS[GTP] hydrolysis.

To further pinpoint the mechanism by which ADAP1-induced PKCθ- and KRAS-dependent ERK phosphorylation, and because ADAP1 has a GAP domain (Fig. 1g), we predicted ADAP1 functions as a KRAS GAP. Supporting this notion, we first confirmed recombinantly purified ADAP1 and KRAS interacted in vitro ($EC_{50} = 288\,nM$) using an Amplified Luminescent Proximity Homogeneous Assay (Alpha) (Fig. 7a, b), which allowed the detection and titration of transient protein–protein interactions. Importantly, ADAP1 also stimulated GTP-to-GDP hydrolysis rate ($K_{obs} = 4.163 \times 10^{-4}\,s^{-1}$) in vitro compared to KRAS intrinsic activity ($K_{obs} = 1.571 \times 10^{-6}\,s^{-1}$) (Fig. 7c, d), signifying ADAP1 is a functional, previously overlooked KRAS GAP.

To understand the physiological significance of ADAP1's in vitro binding and GAP function on KRAS, we next examined ADAP1-KRAS protein–protein interactions in primary T cells. Using a proximity ligation assay we confirmed ADAP1 and KRAS are in physical proximity in stimulated (60 min) $T_M$ (Fig. 7e, left

panel), consistent with the idea of a functional ADAP1-KRAS axis upon T cell activation. Further, TCR/CD28-stimulated $ADAP1^{CRISPR}$ $T_M$ from two donors had reduced $RAS^{GTP}$ levels (Fig. 7f), in agreement with reduced ERK phosphorylation in $ADAP1$ depleted donors ex vivo (Fig. 6f and Supplementary Fig. 6a–c). However, intriguingly, ADAP1 and KRAS were also in proximity in resting $T_M$ prior to stimulation (Fig. 7e, right panel), perhaps maintaining KRAS either inactive at rest or poised for activation during stimulating events as there were no significant differences in $RAS^{GTP}$ levels prior to stimulation across two donors (Fig. 7f). The collective in vitro and in cell data suggest ADAP1 augments GTP-to-GDP nucleotide conversion for optimal KRAS activation and downstream ERK signaling, as seen with other GTPase activation behaviors which need proper nucleotide cycling for regulation of cellular outcomes[38].

## Loss of *ADAP1* predicts immune disorders and results in reduced T cell proliferation ex vivo.

With loss of *ADAP1* dampening KRAS–ERK–AP-1 signaling leading to reduced gene programs, we expected *ADAP1* loss to elicit T cell phenotypic defects in addition to viral gene expression consequences. Gene-disease association analysis of induced genes that were impaired in $ADAP1^{CRISPR}$ scRNAseq (Fig. 5e and Supplementary Data 3) were enriched in phenotypes such as abnormal leukocyte count ($p \sim 2.9 \times 10^{-7}$)/lymphoma ($p \sim 4.5 \times 10^{-7}$) in humans and abnormal T cell physiology ($p \sim 5 \times 10^{-23}$)/abnormal T cell proliferation ($p \sim 2.4 \times 10^{-18}$) in mice (Supplementary Fig. 7a, b). Therefore, we predicted *ADAP1* loss would dampen T cell proliferation due to reduced cell activation upon stimulation.

To test whether reduced T cell growth was due to suboptimal ADAP1-mediated AP-1 signaling, cells were treated with commonly used T cell agonists, PMA (targeting PKCθ–AP-1) and Ionomycin (activating NFAT by calcium release, which is known to cooperate with AP-1 for T cell gene program activation)[39]. Interestingly, while neither agonist alone induced cell proliferation, $ADAP1^{CRISPR}$ $T_M$ showed reduced cell proliferation phenotype in response to co-stimulation (Supplementary Fig. 7c–e), perhaps highlighting the importance of AP-1 and NFAT in ADAP1-mediated cell proliferation induction. Consistently, TCR/CD28-stimulated $ADAP1^{CRISPR}$ $T_M$ also exhibited statistically significant decreased cell count (Supplementary Fig. 7f), albeit with a less drastic phenotype possibly due to redundant regulation of cell proliferation by other TCR/CD28-mediated TFs such as NF-κB. Collectively, *ADAP1* loss compromised T cell proliferation in response to stimulation, consistent with impaired T cell signaling.

## Discussion

Cell signaling-transcriptional programs modulate essential biological processes and are thus often usurped by viruses to regulate their own fate. Here we leveraged HIV-1's dependence on T cell signaling and transcriptional activation to screen for host factors with undescribed T cell signaling functions that reactivate latent HIV-1 provirus. By employing an unbiased gain-of-function screening approach coupled with a large genome cDNA library, we opened the additional possibility of identifying factors that may not be expressed or have impaired function in acute myeloid leukemia cells but are of physiological significance in normal T cells. Expectedly, we found factors previously reported to facilitate T cell signaling as well as latent HIV-1 reactivation. Additionally, we identified ADAP1 as a previously undescribed latent HIV-1 activating factor with no prior functional links to immune cell signaling-transcriptional responses. Further validating the robustness of our gain-of-function approach, we found ADAP1 is expressed in primary CD4+ T cells but not detectable

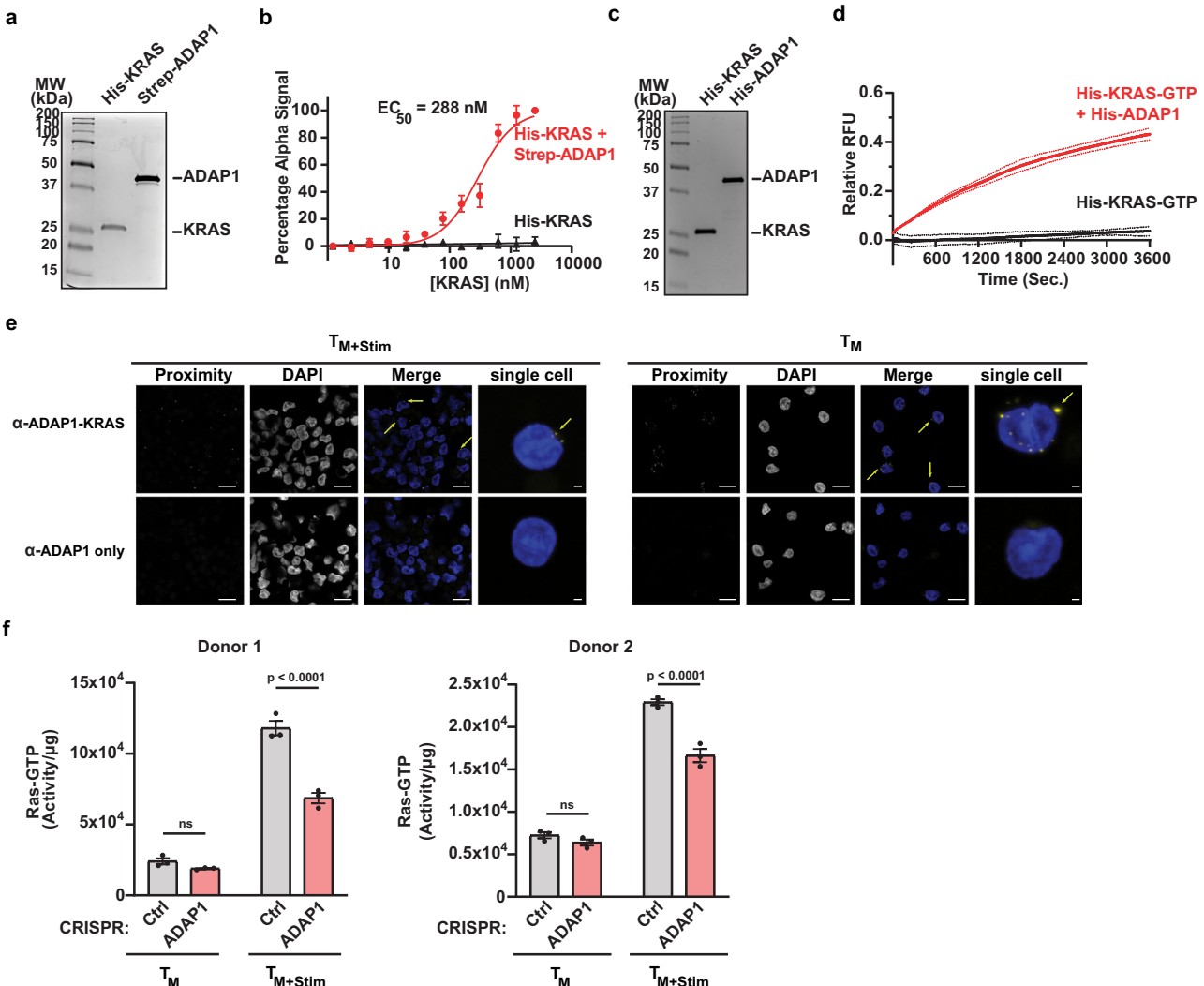

**Fig. 7 ADAP1 interacts with and stimulates KRAS$^{GTP}$ hydrolysis. a** Coomassie staining of recombinant proteins used in Alpha. **b** Alpha confirms binding of recombinant ADAP1 and KRAS. Data represents mean ± s.d. percent ($n = 3$ independent experiments in triplicate). **c** Coomassie staining of recombinant proteins used in GTPase hydrolysis assay. **d** GTP-hydrolysis assay confirms that ADAP1 stimulates the rate of KRAS$^{GTP}$ hydrolysis ($n = 3$). One-phase association fitted to data to calculate $K_{obs}$ hydrolysis rates. Data represents mean ± s.d. Relative RDU ($n = 3$ independent experiments in triplicate). **e** Proximity ligation assay of unstimulated and stimulated (1 h anti-TCR/anti-CD28 beads) primary $T_M$. Proteins were tested for proximity by incubating cells with ADAP1 and KRAS antibodies. As a negative control for non-specific probe ligation or amplification, samples incubated with only one antibody were included. Yellow arrows pointing to yellow puncta indicate proteins are in proximity (distances < 40 nm). Images are representative of two donors. Scale bar is 10 μm for multicell images and 1 μm for single cell images. **f** RAS$^{GTP}$ levels in primary ADAP1$^{CRISPR}$ (red) and Ctrl$^{CRISPR}$ (gray) $T_M$ in the absence or presence of anti-TCR/anti-CD28 bead stimulation (10 min). Each graph is a representative donor. Data represents mean ± s.e.m. of Ras$^{GTP}$/ total protein (μg) ($n = 3$). (two-way ANOVA followed by Sidak's test for multiple comparison between ADAP1$^{CRISPR}$ and Ctrl$^{CRISPR}$ groups), ns not significant. Source data are provided as a Source Data file.

in acute myeloid leukemia cell lines (e.g., Jurkat, K562) (Fig. 1g and https://www.proteinatlas.org/ENSG00000105963-ADAP1/cell+line), potentially explaining why ADAP1 was a hit in the screen. Importantly, loss of *ADAP1* in an ex vivo model of latency dampened latent HIV-1 reactivation (Fig. 3) stressing the physiologic relevance of ADAP1 in tuning HIV-1 gene expression and focusing our efforts to understand ADAP1's role in T cell signaling-gene regulation.

Through a combination of gain-of-function and loss-of-function approaches in cell lines and primary cells, we gained insights into ADAP1 in T cell and HIV-1 biology. We first found ADAP1 interacted with immune signalosome components in stimulated primary T cells, which led us to hypothesize ADAP1 is involved in T cell signaling-transcriptional stimulation during T cell transitions from resting to activated states. Indeed, our

transcriptomics analysis revealed a dampened immune activation in early stimulated ADAP1$^{CRISPR}$ $T_M$ (Fig. 5c–f). One consideration here is that cells were subjected to CRISPR-Cas9 at the $T_E$ state before transitioning to $T_M$ (Fig. 5a); therefore, likely loss of *ADAP1* might have also perturbed the effector-to-memory transition to generate abnormally behaved resting cells with higher activation levels at baseline and lower responsiveness. Supporting this notion, residual levels of Fos (AP-1) were observed in unstimulated ADAP1$^{CRISPR}$ $T_M$ (Supplementary Fig. 5a, b), suggesting ADAP1 is important for the normal effector-to-memory transition. Additionally, in the ex vivo model of latency in where CRISPR-Cas9 was employed at resting state (post-effector-to-memory transition), we noted a slight but statistically significant increase in basal HIV-1 levels with *ADAP1* depletion, but not NF-κB (Supplementary Fig. 3), proposing

ADAP1 may also maintain T cell resting state and HIV-1 latency. Given this finding, further analysis across T cell state differentiation would be of interest to probe from acute vs chronic loss of *ADAP1*, for both HIV-1 and T cell biology.

Because ADAP1 functions at the plasma membrane (Fig. 4a, b) to regulate T cell gene programs (Fig. 5), we predicted *ADAP1* loss would alter a cell signaling-transcriptional axis. We found ADAP1 tunes the KRAS–ERK–AP-1 signaling pathway. Specifically, ADAP1 ectopic expression induced KRAS-dependent ERK phosphorylation in T cell lines (Fig. 6b, d) whereas loss of *ADAP1* reduced ERK phosphorylation in primary T cells (Fig. 6f and Supplementary Fig. 6a–c). We also observed ADAP1 interacts with PKCθ in stimulated cells (Fig. 4f), which others have shown activates the RAS guanine nucleotide exchange factor (GEF) RASGRP1 stimulating RAS–ERK–AP-1 signaling[30]. With targeted stimulation of PKCθ with PMA, we found induced ERK phosphorylation was hampered upon loss of *ADAP1* in primary T cells (Supplementary Fig. 6d). Given both KRAS and PKCθ provoke ADAP1-mediated ERK phosphorylation, these observations raise interesting questions to define the functional interplay between the four factors (ADAP1-PKCθ-RASGRP1-KRAS) during T cell stimulation.

ADAP1 contains an ArfGAP domain, yet perhaps one of the most intriguing findings was evidence indicating ADAP1 interacts with and functions as a KRAS GAP in vitro (Fig. 7a–d). This was supported by proximity of ADAP1 and KRAS in primary T cells (Fig. 7e), and decreased RAS$^{GTP}$ levels in stimulated primary T cells depleted of *ADAP1* (Fig. 7f). Though GAPs are often thought of as negative regulators of GTPase signaling by accelerating GTP-GDP exchange, we observed ADAP1 promoted signaling in terms of increased ERK–AP-1 activation (Fig. 6b). Studies from other GTPases noted GAPs are important for nucleotide cycling to propagate signaling, in where cycling mutants that do not allow nucleotide conversion result in decreased signaling[38]. This interesting observation raises the question as to the purpose of ADAP1 GAP function in the context of T cells. ADAP1 interacts with KRAS at both resting and stimulated states (Fig. 7e), and ADAP1 interacts with PKCθ at stimulated state only (Fig. 4f), which leads to speculation that ADAP1 may have differing roles in resting and

stimulated T cell states. In resting T cells, perhaps ADAP1 promotes GTP-to-GDP conversion off plasma membrane to inactivate KRAS by keeping it in the GDP-bound state (Fig. 8a), a process that avoids cell-intrinsic signaling. In stimulated T cells, ADAP1 binds to KRAS on the plasma membrane where it promotes the continuous cycling of GTP-to-GDP cooperating with an unknown GEF, possibly RASGRP1, to activate KRAS GTPase activity and downstream signaling (Fig. 8b).

We propose that compartmentalized recruitment of KRAS, ADAP1 and a GEF at the plasma membrane facilitates the continuous GTP-to-GDP cycling required for KRAS permanent activation and downstream signaling through the ERK–AP-1 axis. Beyond identifying the KRAS GEF cooperating with ADAP1 in T cell signaling, it will be interesting to determine if the interaction between ADAP1 and PKCθ (and other players in the plasma membrane) alter ADAP1 function regarding KRAS. In neurons, for example, ADAP1 functions as a scaffolding factor for other proteins regardless of their GTP/GDP-bound state, thus opening the possibility that ADAP1 facilitates KRAS recruitment to the immune signalosome. Additionally, it will be interesting to elucidate how and why *ADAP1* expression is further increased in response to T cell stimulation and whether the induced ADAP1 pool plays a distinct function (related or unrelated to KRAS regulation) to the ADAP1 pool normally expressed in memory cells. Answering these questions require further investigation.

Loss of *ADAP1* reduced *IL2* induction (Supplementary Data 3) and cell count in ADAP1$^{CRISPR}$ T$_M$ with targeted co-stimulation of AP-1 and NFAT (Supplementary Fig. 7c). The apparent requirement of both AP-1 and NFAT for ADAP1-regulated cell proliferation are in line with NFAT:AP-1 cooperative transcriptional function to tightly regulate pro-proliferative genes (e.g., *IL2*)[39] and imply that reduced proliferation of ADAP1$^{CRISPR}$ T$_M$ is possibly a result of impaired NFAT:AP-1 cooperativity. Without cooperativity, T cells are unresponsive or become anergic and do not proliferate[40–42], which is in line with the compromised IL2 induction (Supplementary Data 3) and cell proliferation observed in ADAP1$^{CRISPR}$ T$_E$ (Supplementary Fig. 7c–e).

Given IL2 is not only a growth factor, but also stimulates T cell subset-specific cytokines and TFs for proper cell differentiation[43],

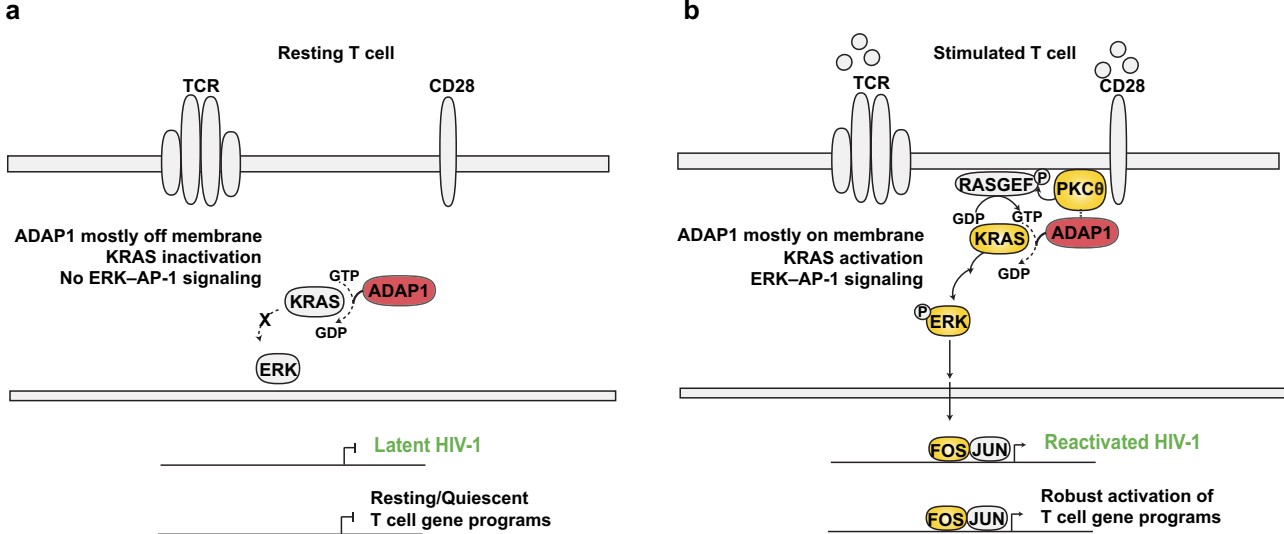

**Fig. 8 Proposed simplified model of ADAP1-mediated T cell signaling stimulation leading to latent HIV-1 reactivation. a** In resting conditions, ADAP1 is largely localized to the cytosol, HIV-1 is in a latent state and T cell programs are quiescent. **b** Upon stimulation (TCR/CD28), ADAP1 is re-localized to the cell membrane where it interacts with immune signalosome proteins including PKCθ and signal transducer protein KRAS, stimulating the MAPK pathway including KRAS–ERK–AP-1 (Fos/Jun). Activated AP-1 binds *cis*-elements on the HIV-1 genome inducing reactivation from latency. For T cell genes, ADAP1-induced AP-1 bind *cis*-elements of genes required for T cell programs, including genes that require NFAT:AP-1 cooperativity.

the observed phenotypes upon *ADAP1* loss raise the interesting possibility that ADAP1 modulates the greater adaptive immune response through the KRAS–ERK–AP-1 axis. Whether such roles in vivo are to tune AP-1 signaling to regulate cell growth during T cell stimulation and differentiation, or immune tolerance, would require currently unavailable mouse models for conditional *ADAP1* ablation in CD4$^+$ T cells.

Because ADAP1 expression activated HIV-1 in our Jkt-HIVLuc model of latency (Fig. 1e, h), it raised the possibility that ADAP1 may enhance current latency reversing agents (LRA). However, treating ADAP1-expressing Jkt-HIVLuc cells with LRAs, JQ1 (a bromodomain inhibitor), Trichostatin A (TSA) a histone deacetylase inhibitor, or suberoylanilide hydroxamic acid (SAHA) another histone deacetylase inhibitor also known as Vorinostat, did not result in synergistic HIV-1 reactivation (Supplementary Fig. 8a–f). These results highlight ADAP1's mechanism of action is different compared to the current LRAs available and motivate developing a compound directly targeting ADAP1. With ADAP1's ability to tune the KRAS–ERK–AP-1 pathway, which is involved in several pathologies including cancer, an ADAP1-specific compound would be worth exploring beyond HIV-1 latency reversal.

While the ex vivo data are consistent with the idea that ADAP1 haploinsufficiency reduces T cell counts in response to immune activation, further work in mouse models is required to provide in vivo relevance. While this mouse model is not yet available, an unchallenged *ADAP1* mutant mouse line (Adap1$^{tm1a(EUCOMM)Wtsi}$ https://www.mousephenotype.org/data/genes/MGI:2442201)[44] displayed increased natural killer (NK) counts but decreased memory markers, potentially linked to altered secretion of IL2 and maybe other cytokines involved in NK maturation, suggesting immune cell proliferation and differentiation irregularities as a consequence of *ADAP1* loss. Interestingly, decreased TCR-mediated T cell proliferation in *PKCθ−/−* mice was attributed to reduced TCR-mediated TF (e.g., AP-1) activity and IL2 production[29], which are in line with the expectation that conditional CD4$^+$ *ADAP1−/−* mice would have impaired AP-1 activation leading to decreased IL2 and potentially decreased T cell proliferation.

The fact that acute myeloid leukemia cells (e.g., Jurkat, K562) do not express detectable ADAP1 (Fig. 1g and The Human Protein Atlas https://www.proteinatlas.org/ENSG00000105963-ADAP1/cell+line) suggests a selective pressure for cell transformation. Consistently, scRNAseq predicted abnormal leukocyte and lymphoma phenotypes upon *ADAP1* loss in stimulated T$_M$ (Supplementary Fig. 7a, b), further suggesting ADAP1 has a role in tuning the identity and magnitude of T cell gene programs to avoid malignancy. It would be interesting to determine whether ADAP1 is important for suppressing tumorigenesis and whether it is through KRAS–ERK–AP-1 regulation. Likewise, the location of *ADAP1* SNPs on potential DNA regulatory elements would require further testing to possibly provide understanding of *ADAP1* expression regulation, potential haploinsufficiency and abnormal T cell proliferation. Overall, these results warrant the study of ADAP1 in an in vivo model yet to exist to appreciate the full extent of ADAP1 influence on T cell fate, immune system defects and tumorigenesis.

## Methods

**Cell lines.** HEK293T (CRL-11268) were obtained from ATCC and maintained in DMEM (HyClone, SH30022.FS) supplemented with 10% fetal bovine serum (FBS) (MilliporeSigma, H9268) and 1% Penicillin/Streptomycin (Pen/Strep) (MP Biomedicals, 091670049). Jurkat CD4$^+$ T cells (TIB-152) were obtained from ATCC and maintained in RPMI-1640 (HyClone, SH30027.FS), 10% FBS, 1% Pen/Strep. Jkt-HIVLuc cells were generated in the D'Orso lab by pNL4.3-delta*Env*-delta*Vpr*-Luc[45] pseudotyped VSVG lentivirus transduction and single cell sorted by the UTSW Flow Cytometry Core into 96-well plates. Clones were expanded and

screened for low basal luciferase activity and high range of inducibility upon TNFα (Sigma, T6674) dose-response stimulation.

**Cloning ORF library.** The 17,384 Ultimate™ ORF LITE clones (Human collection) were obtained from Thermo Fisher scientific. The clones are provided in a Gateway entry vector, pENTR(tm) 221, allowing rapidly and efficiently transferring of the ORF of interest to any expression (Gateway destination) vector. The collection was divided in 183 minipools of 96 hORFs (i.e., one 96-well plate of the collection). For each minipool, the hORFs were cloned *en masse* from the pENTR(tm)221 into the lentiviral expression vector pTRIP.CMV.IVSB.IRES.TagRFP-DEST[46] by Gateway LR reaction (the recombination between the attL (on entry clone) and attR (on destination vector) sites to generate an expression clone). After electroporation, transformants were selected on Luria Broth (LB) plus ampicillin and the lentiviral vector was extracted. Lentiviruses for each minipool were generated in 293T cells by co-transfection of plasmids expressing: pTRIP.CMV.IVSB.minipool.IRES.TagRFP, HIV-1 gag-pol, and the vesicular stomatitis virus glycoprotein (VSVG) in a ratio of 1:0.8:0.2. Supernatants were collected at 48 and 72 h post-transfection, pooled, cleared by centrifugation at $1000 \times g$, aliquoted and stored at −80 °C.

**Gain-of-function screen approach.** The gain-of-function screen approach outlined in Fig. 1c schematic, was performed using a 3-phase strategy (pool 72 is highlighted as example). Phase 1: a human ORF cDNA library ($n = 17,384$ cDNAs) arrayed in 96-well plates ($n = 183$ plates) were combined such that each plate formed a pool ($n = 183$ pools each containing 96 cDNAs). The cDNA pools were used to generate pooled lentiviruses and transduce Jkt-HIVLuc cells followed by evaluating latent HIV-1 reactivation using luciferase assays (see Fig. 1d and Supplementary Fig. 1c for results). Phase 2: positive pools (selected using criteria in Supplementary Fig. 1d) were resolved by re-pooling the original 96-well plate into smaller pools. To reduce false positives due to the combinatorial action of two or more factors within the same pool, we reconsolidated each 96-well plate so that each row (A-H, 12 cDNAs each) and each column (1-12, 8 cDNAs each) formed individual pools, thus allowing each individual cDNA to be screened twice (see Supplementary Fig. 1e for example). Phase 3: positive row (e.g., 72F) and column pools (e.g., 72.3) were mapped back to the original plate to identify and experimentally validate intersecting wells (e.g., 72F3) as the latent HIV-1 activating host factor (see Supplementary Fig. 1f for example results).

**Lentivirus production and transduction of Jkt-HIVLuc cells for luciferase assays.** To generate lentivirus, HEK293T plated in a 6-well plate were transfected with 1 µg pTRIP-cDNA vector, 1 µg gag/pol (psPAX2, Addgene, 12260), and 0.25 µg VSVG (pMD2.G, Addgene, 12259) using 3 µl Polyjet (SignaGen, SL100688) per well. Cell supernatants were collected and cleared of cell debri by using 0.45 µm syringe filters (Millipore, SLHV033RS) 48 h post-transfection. Viral transduction was done by spinoculation using $1 \times 10^5$ Jkt-HIVLuc cells, 8 µg/ml polybrene (MilliporeSigma, H9268), and RPMI-1640 to a final volume of 0.1 ml per well of a flat-bottom 96-well plate at room temperature for 2 h at $400 \times g$ and later crowding for 4 h at 37 °C. Cells were spun down, virus was removed, and cells were incubated in 0.2 ml RPMI-1640/10% FBS/1% Pen/Strep for 48 h. After 48 h, cells transduction efficiency was measured by flow cytometry (A600 HTAS, Stratedigm) by detecting red fluorescent protein (RFP) expression. Firefly luciferase reporter activity was measured by incubating ~$2 \times 10^5$ cells in 25 µl lysis buffer (Promega, E1531) for 5 min while orbital shaking and using 20 µl of lysate with 20 µl of Luciferase Kit substrate (Promega, E1501).

**Western blot assays.** Total protein extracts from $1 \times 10^5$ cells were electrophoresed on home-made 8 or 10% polyacrylamide SDS-PAGE gels using 1X Tris-Glycine-SDS running buffer. Gels were transferred onto 0.45 µm nitrocellulose membranes (Bio-Rad, 1620115) using Bio-Rad Trans-Blot Turbo Transfer System, blocked for 1 h in 5% Milk + Tris-buffered saline-Tween-20 (TBS-T) for most targets or 5% BSA + TBS-T for phosphorylated targets. Membranes were probed with primary antibody in appropriate blocking buffer overnight at 4 °C. Membranes were washed three times with TBS-T for 10 min at room temperature and probed with secondary antibody in the same blocking buffer used for primary antibody at 1:10,000 dilution for 60 min at room temperature. Blots were washed as before and incubated with Clarity Western ECL (Bio-Rad, 1705061) for 5 min or Super Signal West Femto (Thermo Fisher, 185022) for 10 s. Images were acquired using the Chemidoc Imaging System (Bio-Rad). Signal intensity quantification of bands was done using Bio-Rad ImageLab software.

**Antibodies.** Anti-Centaurin alpha1 G-4 (ADAP1) (Santa Cruz, sc-390498) (1:1000); Anti-p65/NF-κB (Santa Cruz, sc-372) (1:4000); Anti-phos-ERK1/2 Thr202/Tyr204 (Cell Signaling Technology, 4370) (1:1000); Anti-ERK1/2 (Cell Signaling Technology, 4696 or 4695) (1:2000); Anti-Fos (Cell Signaling Technology, 2250) (1:1000); Anti-Jun (Cell Signaling Technology, 9165) (1:1000); Anti-Flag M2 (Sigma, F31165) (1:10,000); Anti-PKCθ (Cell Signaling Technology, 13643) (1:5000); Anti-CD3ε (Cell Signaling Technology, 4443) (1:5000); Anti-GAPDH (Cell Signaling Technology, 2118) (1:5000); hFAB Rhodamine anti-Actin (Bio-Rad, 12004166); (1:10,000); Anti-StrepTactin-HRP (Bio-Rad, 161-0381) (1:10,000); Anti-rabbit IRDye 800CW (Licor, 926-32211) (1:10,000); Anti-mouse IRDye

680CW (Licor, 925-68072) (1:10,000); Anti-mouse HRP (Cell Signaling Technology, 7076) (1:10,000); Anti-rabbit HRP (Cell Signaling Technology, 7074), (1:10,000); Anti-rat HRP (Abcam, ab97057) (1:10,000).

**Site-directed mutagenesis**. ADAP1 and KRAS point mutants were generated using QuikChange II XL Site-Directed Mutagenesis kit (Agilent, 200522) per manufacturer's instructions. Briefly, mutagenesis primers were designed using the Agilent QuikChange Primer Design online tool (https://www.agilent.com/store/primerDesignProgram.jsp). PCR amplified DNAs were transformed into *E. coli* BME treated XL-10 Gold provided in the kit and positive clones validated by Sanger Sequencing at the UTSW McDermott Sanger Sequencing Core. The primers used are as follows:

GAP Mut C24A (Fwd 5′-CGCGCGCTGCGCGGACGCCGGCGCCCCGGAT CC-3′; and Rev 5′-GGATCCGGGGCGCCGGCGTCCGCGCAGCGCGCG-3′).

PH1 Mut R149C (Fwd 5′-CAACGGGCAGTTTTTGAGCTGCAAGTTTGTGC TGACAGAAC-3′; and Rev 5′-GTTCTGTCAGCACAAACTTGCAGCTCAAAAA CTGCCCGTTG-3′).

KRAS S17N (Fwd 5′-CTGGTGGCGTAGGCAAGAATGCCTTGACGA-3′; and Rev 5′-TCGTCAAGGCATTCTTGCCTACGCCACCAG-3′).

**Isolation of naïve CD4+ T cells and generation of T$_M$**. Leukopaks (~50 ml) from random and anonymous healthy donor blood samples were purchased from Gulf Coast Regional Blood Center (Houston, Tx) and used in this study in accordance to local ethical standards. To generate T$_M$, PBMCs were isolated from 50 ml leukopaks of healthy donor blood samples (Gulf Coast Regional Blood Center) using Lymphoprep density gradient (STEMCELL, 07801). T$_N$ were isolated from PBMCs following the instruction of EasySep™ Human Naïve CD4+ T Cell Isolation Kit (STEMCELL, 558521). Purity was analyzed by flow cytometry comparing the presence of well-defined markers to isotype controls: CD4-PerCP-Cy5.5 (eBioscience, 45-0049-42) 0.125 µg/test, control: mouse IgG1k isotype PerCP-Cy5.5 (eBioscience, 45-4714-82) 0.125 µg/test, CD3-PE/Cy7 (BioLegend, 300419) 0.125 µg/test, control: mouse IgG1k isotype PE/Cy7 (Invitrogen, 25-4714-80) 0.125 µg/test; CD45RO-PE (eBioscience, 12-0457-41) 0.125 µg/test, control: mouse IgG2a isotype PE (eBioscience, 12-4724-42) 0.125 µg/test; CD45RA-FITC (eBioscience, 11-0458-42) 0.125 µg/test, CD25-FITC (eBioscience, 11-0257-42) 0.125 µg/test, control: mouse IgG2bκ isotype FITC (eBioscience, 11-4732-41) 0.125 µg/test. Cells were then activated by culturing with 1 µl anti-IL-4 (PeproTech, 500-P24), 2 µg/ml anti-IL-12 (PeproTech, P154G), 10 ng/ml TGF-β1 (PeproTech, 100-21) and Dynabeads human T-activator anti-TCR/anti-CD28 (Gibco, 11132) (1 bead/cell). After 3 days, beads were removed by column-free magnetic separation. Cells were cultured in complete media with 30 IU/ml IL2 (Roche, 202-IL). Media containing IL2 was replenished daily for 10 days, followed by every other day, and maintaining cells at 1 × 10$^6$/ml. Transition from T$_E$ into T$_M$ was monitored by flow cytometry analysis of intracellular Ki67 (eBioscience, 12-5699-41) 0.125 µg/test, control: mouse IgG1κ Isotype PE (eBioscience, 12-4714-41) 0.125 µg/test and surface CD45RA-FITC/CD45RO-PE, in where low expression of Ki67 (<5%) were considered low dividing and therefore evidence of quiescence.

**Flow cytometry analysis**. All samples were prepared by transferring 2 × 10$^5$ cells per sample to an uncoated V-bottom 96-well plate (Nunc). The samples were spun down at 300 × *g* for 5 min at room temperature and washed with 0.2 ml 1X PBS twice. Cells were stained according to target protein. Samples were run on a 96-well plate reader (A600 HTAS, Stratedigm) with a gate cell count set at 20,000 events using CellCapTure flow cytometry software (Stratedigm). Data analysis was performed with FlowJo version 10.1.

For the detection of RFP, cells were fixed with 1% paraformaldehyde (PFA) for 5 min at room temperature. The PFA was washed twice with 0.2 ml of 1X PBS before resuspending in 0.1 ml 1X PBS/2% FBS and immediately analyzing by flow cytometry.

For the detection of cell surface proteins, cells were resuspended in 0.1 ml staining buffer (1X PBS/2% FBS) and incubated for 20 min on ice. Cells were then stained with antibodies against target or appropriate isotype control according to manufacturer's protocol and incubated on ice protected from light for 30 min. After incubation, cells were washed twice in staining buffer, resuspended in 0.15 ml staining buffer, fixed as stated above and analyzed immediately by flow cytometry.

For the detection of intracellular proteins, cells were fixed as stated above and permeabilized using 1X BD Perm/Wash (554723) for 15 min at room temperature and then stained with appropriate antibody or isotype control and incubated for 30 min at 4 °C protected from light. Cells were washed twice with 1X BD Perm/Wash before being resuspended in staining buffer and were immediately analyzed by flow cytometry.

**Immunoprecipitation and mass spectrometry**. T$_M$, resting or stimulated for 1 h with Dynabeads human T-activator anti-TCR/anti-CD28 (Gibco, 11132) (1 bead/cell), were IP by Dynabeads protein G (Invitrogen, 10004D) using mouse anti-ADAP1 (Sigma, SAB1408812) or Normal Mouse IgG (Millipore, 12–371). In total, 10% of the elutions were saved to verify IP efficiency by western blot. 50% of the elutions were electrophoresed on an SDS-PAGE pre-cast gel (Bio-Rad, 456–1033) and stopped when sample ran 10 mm into resolving gel. The gel was Coomassie

stained, and ~10 mm gel slices were excised and diced before submitting to the UTSW Proteomics Core. Samples were digested overnight with trypsin (Pierce) following reduction and alkylation with DTT and iodoacetamide (Sigma Aldrich). The samples then underwent solid-phase extraction cleanup with an Oasis HLB plate (Waters), and the resulting samples were injected onto an Orbitrap Fusion Lumos mass spectrometer coupled to an Ultimate 3000 RSLC-Nano liquid chromatography system. Samples were injected onto a 75 mm i.d., 75-cm long Easy-Spray column (Thermo) and eluted with a gradient from 0–28% buffer B over 90 min. Buffer A contained 2% (v/v) acetonitrile and 0.1% formic acid in water, and buffer B contained 80% (v/v) acetonitrile, 10% (v/v) trifluoroethanol, and 0.1% formic acid in water. The mass spectrometer operated in positive ion mode. Mass spectrometry scans were acquired at 120,000 resolution in the Orbitrap and up to 10 MS/MS spectra were obtained in the ion trap for each full spectrum acquired using higher-energy collisional dissociation for ions with charges 2–7. Dynamic exclusion was set for 25 s after an ion was selected for fragmentation. Raw MS data files were analyzed using Proteome Discoverer v2.4 SP1 (Thermo Fisher Scientific), with peptide identification performed using Sequest HT searching against the human protein database from UniProt. Fragment and precursor tolerances of 10 ppm and 0.6 Da were specified, and three missed cleavages were allowed. Carbamidomethylation of Cys was set as a fixed modification, with oxidation of Met set as a variable modification. The false-discovery rate cutoff was 1% for all peptides. High confidence interactors with a linear Fc >2 compared to IgG control were subjected to GO analysis using Metascape (http://metascape.org)[47] and generated networks were uploaded into, and visualized using Cytoscape version 3.8.2 (https://cytoscape.org/)[48].

**10x single cell RNA sequencing**. ADAP1$^{CRISPR}$ and Ctrl$^{CRISPR}$ T$_M$ were left resting or stimulated with Dynabeads human T-activator anti-TCR/anti-CD28 (Gibco, 11132) (1 bead/cell) for 4 h at 37 °C. Beads were removed by column-free magnetic separation and resuspended in RPMI-1640. The four samples were submitted to the UTSW McDermott Center Sequencing Core where samples were prepared for scRNAseq using the 10X Genomics Next GEM Single cell 3′ Reagent Kit v3.1. Single cell suspensions were washed in 1X PBS (calcium and magnesium free) containing 0.04% weight/volume BSA (400 µg/ml) and brought to a concentration of ~700–1200 cells/µl. This was accomplished by staining 10 µl of the single cell suspension with Trypan blue and reading on the Countess™ II Automated Cell Counter (Thermo Fisher). The appropriate volume of cells was loaded with Single Cell 3' Gel Beads into a Next GEM Chip G and run on the Chromium™ Controller. Gel Bead-In Emulsions (GEM) emulsions were incubated and then broken. Silane magnetic beads were used to clean up GEM reaction mixture. Read one primer sequence was added during incubation and full-length, barcoded cDNA was then amplified by PCR after cleanup. Sample size was checked on a Tapestation 4200 (Agilent) using the DNAHS 5000 tape and concentration determined by the Qubit 4.0 Fluorimeter (Thermo Fisher) using the DNA HS assay. Samples were enzymatically fragmented and underwent size selection before proceeding to library construction. During library preparation, read 2 primer sequence, sample index and both Illumina adapter sequences were added. Subsequently, samples were cleaned up using Ampure XP beads (Beckman Coulter) and post library preparation quality control was performed using the DNA 1000 tape on the Agilent Tapestation 4200. Final concentration was ascertained using the Qubit DNA HS assay. Samples are loaded at 1.6 pM and run on the Illumina NextSeq500 High Output FlowCell using V2.5 chemistry. Run configuration is 28 × 98 × 8.

**scRNAseq analysis**. Four different conditions (Ctrl_0hr, Ctrl_4hr, KD_0hr, and KD_4hr) of T$_M$ were loaded into the 10X Genomics chromium controller. The 10X Genomics' analysis pipeline, cellranger (version 3.0.0), was used to demultiplex chromium data, filter out low quality cell barcodes, and produce a gene-cell matrix. Here, transcriptomes were aligned to GRCh38 using STAR (version 2.5.1b). On R (version 4.0.2), Seurat (version 4.0.0), an R package for single cell transcriptomics[49], was used to create Seurat objects from gene-cell matrices and process each Seurat object, and later to integrate multiple Seurat objects together. In detail, low quality cells and multiplets were filtered out using stringent thresholds (cells with higher than 90 percentile of no. features or no. feature counts or with lower than 10 percentile of no. features or no. feature counts were removed). Also, cells with percentages of mitochondrial genes higher than 10% were removed. To address inherent issues in single cell transcriptomics, such as technical noise from gene sampling fluctuations and cell-to-cell variations in sequencing efficiency, a pre-processing function based on a regularized negative binomial regression, SCTranform[50], was used to normalize and stabilize the technical noise variance of UMI counts. Then, 7500 features variable in objects to integrate were selected and, using PrepSCTIntegration, residuals for the selected features were ensured to be present in each of the objects. Using so-called anchors, pairs of cells from each object that were identified to have mutual nearest neighbors in a dimensionality-reduced space, the objects were finally transformed into a shared space, achieving a data integration[51]. Cell clusters in an integrated dataset were identified by a graph-based clustering method (resolution = 0.75) and were visualized using UMAP, a nonlinear dimensional reduction technique. To see in which cluster expression of the activated cells are enriched, select markers (CD69, MYC, GZMB, LTA, CD40LG, RGCC, NR4A1, TNFα, IL2, and CSF2) were compared to randomly selected control genes, so-called active cell scores were

calculated using *AddModuleScore* and then visualized over UMAP plots. In addition, for an integrated object of Ctrl_0hr and Ctrl_4hr objects, DEGs were searched using a Wilcoxon Rank Sum test in its normalized RNA assay at a $Log_2FC$ threshold = 0.25 (corresponding to about 1.19-fold), resulting in 882 Ctrl DEGs. DEGs between KD_0hr and KD_4hr objects were also searched in the same manner and 537 KD DEGs were found. To identify genes showing different levels of fold changes caused by the cell activation treatment between Ctrl and KD conditions, $log_2FC$ value differences between the conditions for the Ctrl and KD DEGs were explored and depicted in a scatter plot.

**Transcription reporter assays.** Strep-tagged ADAP1 or GFP were expressed using the Jurkat T-REx system. Briefly, the Jurkat T-REx cells (Thermo Fisher) were electroporated with 1 μg of Ssp1-linearized pcDNA4/TO Strep-tagged ADAP1 or GFP vector using Nucleofector2b (Lonza). Positive clones were selected with 10 μg/ml Blasticidin and 100 μg/ml Zeocin. To induce protein expression, the stable cell lines were treated with 1 μg/ml Doxycycline Hydrochloride (Fisher Scientific, BP2653-1) for 48 h. Next, for each transcription reporter assay reaction, $1 \times 10^6$ Jurkat cells expressing Strep-tagged ADAP1 or GFP were resuspended in 0.1 ml Mirusbio ingenio (Mirus, MIR50115) solution containing 2 μg of TF-responsive reporters driving Firefly luciferase (pGL3-3xAP1-luciferase (Addgene, 40342), pGL3-NF-κB-luciferase (Promega), pGL3-NFAT-luciferase (Addgene, 17870)) and 0.2 μg Renilla plasmid (pRL-CMV, Promega, E2261) at room temperature. Triplicate reactions were set up for each reporter plasmid. Cells were transferred into cuvettes (2 mm gap, Mirus, MIR50121) and subjected for electroporation using Nucleofector2b (Lonza). Cells were transferred to pre-warmed plate containing 1 ml RPMI/10% FBS and cultured for 24 h at 37 °C. Cells were harvested and analyzed using Dual Luciferase Kit (Promega, E1910). Firefly luciferase signal was normalized to the internal Renilla luciferase control.

**Expression and purification of recombinant Strep-tagged proteins from bacterial sources.** For Strep-tagged ADAP1 used in Fig. 7a, pET30a plasmid DNAs were chemically transformed into *E. coli* (BL21 DE3) (NEB, C2527I) and grown on LB agar plates containing 50 μg/ml kanamycin overnight at 37 °C. A single colony was used for inoculating 3 ml of LB + 50 μg/ml kanamycin overnight. The following day, 500 ml of LB + 50 μg/ml kanamycin was inoculated by the small culture at 1:50 dilution and grown at 37 °C to an $OD_{600nm} = 0.6$. Protein expression was induced by adding a final concentration of 0.1 mM IPTG and incubating at 18 °C for 16 h. Cultures were pelleted by centrifugation and resuspended in lysis buffer (20 mM Tris-HCl pH = 8.0, 150 mM NaCl, 10% glycerol, 1 mM DTT, 1 mM PMSF, EDTA-free protease inhibitors (Roche, 4693159001), 1 mg/ml lysozyme (Fisher, BP535-1)) for 15 min at 4 °C. Cells were subjected to further disruption by sonication, then centrifuged $20,000 \times g$ for 15 min at 4 °C and the supernatant was further cleared by passing through a 0.45 μm filter.

Proteins were purified by gravity flow using StrepTactin® Superflow® high-capacity resin (IBA, 2-1208-010) in 2 ml chromatography columns (Bio-Rad, 7311550) and washed twice with 10 ml column volumes with wash buffer (20 mM Tris-HCl pH = 8.0, 150 mM NaCl, 10% glycerol, 1 mM DTT, 0.5% Triton X-100). Proteins were eluted off the column using buffer containing 2.5 mM D-biotin. After purification, the elution was concentrated using 10 kDa Amicon Ultra-0.5 Centrifugal Filter Units (Millipore, UFC505024) into buffer (20 mM Tris-HCl pH = 8.0, 150 mM NaCl, 10% glycerol). Protein concentration was approximated by Coomassie staining alongside BSA standards.

**Expression and purification of recombinant 6xHis-tagged proteins from bacterial sources.** For His-tagged proteins used in Fig. 7c, pET30a plasmid DNAs were chemically transformed into *E. coli* (BL21 DE3) (NEB, C2527I) and grown on LB agar plates containing 50 μg/ml kanamycin overnight at 37 °C. A single colony was used for inoculating 3 ml of LB + 50 μg/ml kanamycin overnight. The following day, 500 ml of LB + 50 μg/ml kanamycin was inoculated by the small culture at 1:50 dilution and grown at 37 °C to an $OD_{600nm} = 0.6$. Protein expression was induced by adding a final concentration of 0.1 mM IPTG and incubating at 18 °C for 16 h. Cultures were pelleted by centrifugation and resuspended in lysis buffer (20 mM Tris-HCl pH = 8.0, 150 mM NaCl, 10% glycerol, 1 mM DTT, 1 mM PMSF, EDTA-free protease inhibitors (Roche, 4693159001), 1 mg/ml lysozyme (Fisher, BP535-1)) for 15 min at 4 °C. Cells were subjected to further disruption by sonication, then centrifuged $20,000 \times g$ for 15 min at 4 °C and the supernatant was further cleared by passing through a 0.45 μm filter.

Proteins were purified by gravity flow using Ni-NTA agarose (Qiagen, 30210) in 2 ml chromatography columns (Bio-Rad, 7311550) and washed with 10 ml column volumes twice with low imidazole containing wash buffer (20 mM Tris-HCl pH = 8.0, 150 mM NaCl, 10% glycerol, 1 mM DTT, 0.5% Triton X-100, 20 mM imidazole). Proteins were eluted off the column in a stepwise manner with wash buffer containing 50 mM and 200 mM Imidazole (three times each buffer). After purification, fractions were electrophoresed on SDS-PAGE, and fractions containing protein of the appropriate size were pooled and concentrated using 10 kDa Amicon Ultra-0.5 Centrifugal Filter Units (Millipore, UFC505024) into buffer without imidazole (20 mM Tris-HCl pH = 8.0, 150 mM NaCl, 10% glycerol). Protein concentration was approximated by Coomassie staining alongside BSA standards.

**GTPase hydrolysis assay.** KRAS was loaded with GTP nucleotide by incubating 100 μM protein with 10 mM EDTA and 500 μM ultra-pure GTP (Thermo Scientific, R0461) for 2 h at room temperature with slow shaking (150 rpm). Unbound GTP was removed using 2 ml Zeba column with 7 kDa molecular weight cutoff (Thermo Scientific, 89889) and exchanged into 30 mM Tris pH = 7.5 buffer containing 1 mM DTT. The protein sample was immediately used in GTP hydrolysis in vitro assays using EnzChek Phosphate Assay Kit (Invitrogen, E6646) following manufacturer instructions. Reactions were performed in 384-well clear microplates with 30 mM Tris pH = 7.5 buffer containing 1 mM DTT. For each sample reaction, final concentration of the following reagents was added: 200 μM MESG, 1 U/ml PNP, 50 μM $KRAS^{GTP}$. To start reaction, ADAP1 (25 μM final concentration) plus 40 mM $MgCl_2$ or buffer plus 40 mM $MgCl_2$ were added quickly and measured absorbance at 360 nm every 4 s for 1 h. Data were analyzed by subtracting the background (no-substrate control) at each measurement and graphing relative light units using GraphPad 9.0.2. Data were fit to a one-phase association curve to obtain relative GTP-hydrolysis rate ($K_{obs}$).

**Amplified luminescent proximity homogenous assay (Alpha).** In total, 500 nM Strep-tagged ADAP1 was combined with a range of His-tagged KRAS in reaction buffer consisting of 20 mM HEPES pH = 7.5, 100 mM NaCl, 5% glycerol, in a 384-well plate. Plates were sealed and shaken, then incubated for 2 h at room temperature. StrepTactin® Alpha donor beads (PerkinElmer, AS106D) and nickel chelate (Ni-NTA) Alpha acceptor beads (PerkinElmer, 6760619C) were pre-mixed and added to the reaction at a final concentration of 10 μg/ml. Following an overnight incubation at 4 °C, Alpha signal was measured using a Synergy Neo plate reader (BioTek) with the standard vendor recommended AlphaScreen settings. GraphPad Prism software (version 9.0.2) was used to analyze data. The $EC_{50}$ values were determined by nonlinear regression of plots of [KRAS] vs. percentage of alpha signal.

**Proximity ligation assay (PLA).** To determine whether endogenous protein–protein interaction between ADAP1 and KRAS exists within unstimulated and stimulated $T_M$, Duolink Proximity Ligation Assay (Sigma, DUO92101) was performed to detect proximity of proteins up to 40 nm. Protocol was followed per manufacturer's instructions. Briefly, for each treatment, $1 \times 10^6$ $T_M$ were resuspended in 1 ml RPMI media and either left unstimulated or stimulated with Dynabeads human T-activator anti-TCR/anti-CD28 (Gibco, 11132) (1 bead/cell) for 60 min at 37 °C. Samples were fixed in 1% PFA for 5 min at room temperature and washed 3 times with 1X PBS. Cells were resuspended in 0.1 ml PBS. In total, 10–20 μl of cell suspension was spread onto pre-cleaned fluorescent antibody slides (Thermo Fisher, 3032-002) coated with poly-L-lysine (Sigma, P8920) and allowed to slightly dry for adherence. Cells were permeabilized by incubating with 0.5% Triton X-100 in PBS for ~1–2 min at room temperature followed by washing twice with 1X PBS.

Cells were blocked using provided blocking solution for 60 min at 37 °C in home-made humidity chamber. Slides were then incubated with mouse anti-ADAP1 (Sigma, SAB1408812) and rabbit-anti-KRAS (Abcam, ab172949) primary antibodies in provided antibody diluent and incubated overnight at 4 °C in humidity chamber. For negative controls, samples were incubated with only one or none of the primary antibodies. The next day, slides were washed with wash buffer A provided and incubated with antibody diluent containing PLA anti-mouse and anti-rabbit probes provided for 1 h at 37 °C. Slides were washed as before and incubated with provided ligation buffer containing ligase for 30 min at 37 °C. Slides were washed as before and incubated with amplification buffer containing polymerase for 100 min at 37 °C. Slides were washed in wash buffer B provided in the kit. Provided in situ mounting media with DAPI was added to cells and mounted with a cover slip and sealed with nail polish. Samples were analyzed using Confocal/Multiphoton Zeiss LSM880 Inverted microscope with ×63 oil immersion objective at UTSW Live Cell Imagine Core. Single primary antibody and no antibody samples were used to set laser power and gain for each channel (DAPI-405 laser, PLA-594 laser) and maintained consistent across all samples. Z-stacks were obtained for each sample. Images were analyzed using FIJI ImageJ 1.53c. For Fig. 7e resting cells, single slice images were converted to grayscale, Look-up Tables (LUT) were maintained consistent across all samples. Images were flattened and exported as TIFF. For Fig. 7e stimulated cells, Z-stack with 2.74-micron depth were used to generate maximum projection images. LUT were maintained consistent across all samples and through Z-stack. Composite image was generated, flattened, and exported as TIFF.

**$RAS^{GTP}$ ELISA.** For the assays in Fig. 7f, $RAS^{GTP}$ levels in cells were quantified using $RAS^{GTP}$ ELISA kit (Active Motif, 52097) per manufacturer's protocol. Briefly, GST-Raf RAS Binding Domain was coated onto provided plate for 1 h at 4 °C. During incubation, cell samples were prepared in triplicate. $T_M$ cells were resuspended in RPMI at $2 \times 10^6/0.1$ ml per sample. To each sample, 0.1 ml media or media containing Dynabeads human T-activator anti-TCR/anti-CD28 (Gibco, 11132) (1 bead/cell) were added to sample and incubated at 37 °C for 10 min. Reactions were stopped by adding ice cold PBS and pelleted at $300 \times g$ for 5 min at 4 °C. Supernatants were immediately removed, cells lysed in provided lysis buffer and incubated on ice for 15 min. Lysate was cleared by centrifugation for 10 min at

20,000 × g at 4 °C. An aliquot of lysate was saved for quantifying total protein using BCA assay (Pierce, 23227). ELISA plate was washed three times with provided wash buffer and cleared lysate was added to plate, along with provided positive control and blank wells. Plate was incubated 1 h at room temperature, washed three times as before, and incubated with provided anti-RAS antibody for 1 h at room temperature. Process was repeated for secondary HRP antibody and samples were detected by chemiluminescent detection on luminometer plate reader. Amount of $RAS^{GTP}$ was quantified as activity/μg.

**Immunofluorescence**. For images in Fig. 4a, cells were fixed with 1% PFA for 5 min in 96 U-bottom non-TC coated plates. After fixation, cells were pelleted and washed three times with 1X PBS. Samples were permeabilized by treating with 0.5% Triton X-100 for 2 min at room temperature and washed three times with 1X PBS. Blocking was done by incubating samples with blocking buffer (5% normal goat serum, 4% bovine serum albumin, 1X PBS) for 1 h at room temperature. Samples were stained with primary antibody (mouse anti-ADAP1 (Sigma, SAB1408812)) in blocking buffer overnight at 4 °C. The next day samples were washed three times with 1X PBS and incubated with secondary antibody (goat anti-mouse IgG H + L Alexa Fluor 594 (Invitrogen, A-11032)) diluted in blocking buffer for 1 h at room temperature in the dark. Samples were stained for DAPI diluted in water for 2 min and washed three times with water before drying cells on poly-L-lysine (Sigma, P8920) coated coverslips at room temperature protected from light. Coverslips were mounted onto pre-cleaned fluorescent antibody glass slides (Thermo Fisher, 3032-002) using ProLong Gold Antifade Mountant (Invitrogen, P36930) and cured for 24 h protected from light before imaging with Confocal/Multiphoton Zeiss LSM880 Inverted microscope with ×63 oil immersion objective at UTSW Live Cell Imagine Core. Z-stacked for a total depth of 1.43 μm images were taken and used to generate maximum projection images with FIJI ImageJ 1.53c. Mean fluorescence of negative secondary antibody-only stained samples were subtracted from all sample images. Single cell image is a zoom in of larger panel indicated by dashed square.

**Subcellular fractionation**. $T_M$ cells were either unstimulated or stimulated with Dynabeads human T-activator anti-TCR/anti-CD28 (Gibco, 11132) (1 bead/cell). Following stimulation, $5 \times 10^6$ cells were subjected to fractionation as follows. Cells were washed with 1X PBS and resuspended in 500 μl of fractionation buffer pH 7.4 (25 mM Hepes, 250 mM sucrose, 1 mM MgCl$_2$, 2 mM EGTA, EDTA-free protease inhibitors (Roche, 4693159001)). Input samples of 50 μl (10%) were reserved. The remaining cells were lysed by 4 rounds of liquid nitrogen freeze-thaw plus 5 s of sonication at 75% amplitude on a water bath sonicator (Qsonica Q800R3). Samples were then subjected to ultracentrifugation at 103,000 × g for 30 min at 4 °C (Beckman Coulter Optima Max). Supernatants were removed and cell pellets washed 4 times with cell fractionation buffer to remove residual supernatant. Samples were prepared for SDS-PAGE and western blotting. GAPDH was used as a cytosolic marker and CD3ε was used as an integral membrane marker.

**CRISPR-Cas9 genomic editing for generation of ADAP1$^{CRISPR}$ and Ctrl$^{CRISPR}$ $T_M$**. Predesigned guideRNAs (gRNA) were purchased from IDT for targets: Ctrl (IDT, 1072544), ADAP1 (IDT, Hs.Cas9.ADAP1.1.AA), CXCR4 (IDT, Hs.Cas9.CXCR4.1.AA), p65/NF-κB (IDT, Hs.Cas9.RELA.1.AA) as suggested by the manufacturer (https://www.idtdna.com/site/order/designtool/index/CRISPR_PREDESIGN). $T_N$ were isolated and TCR/CD28 stimulated as above for the generation of $T_M$. At day 3 post stimulation , equal molar ratios of tracrRNA, ATTO550 (IDT, 1075928) and gRNA were mixed and heated at 95 °C for 5 min and allowed to cool to room temperature. TrueCut Cas9 Protein v2 (Fisher, A36498) was added at 2.5X molar ratio of tracrRNA/gRNA complex and incubated at room temperature for 20 min. Pre-assembled Cas9-gRNA RNPs were electroporated into TCR/CD28 stimulated cells using Lonza Human T cell Nucleofector kit (Lonza, VPA-1002) and the Nucleofector2b equipment. After 2 days, electroporation efficiency was monitored by measuring fluorescent tracrRNA, ATTO550 incorporation using flow cytometry. Efficiency of target protein depletion was monitored by western blotting.

**Ex vivo model of HIV-1 latency**. VSVG pseudotyped viruses (pNL4.3-delta*Env*-nLuc-2A*Nef* -VSVG) were produced by co-transfecting pNL4.3-delta*Env*-nLuc-2A*Nef* and pCMV-VSVG into HEK293 cells. Cell supernatants were collected and filtered with a 0.22 μm filter after 2 days. Viruses were aliquoted and stored at −80 °C.

$T_N$ were isolated and TCR/CD28 stimulated as above for the generation of $T_M$. At day 3 post stimulation, cells were infected with pseudotyped viruses (pNL4.3-delta*Env*-nLuc-2A*Nef* -VSVG) using spinoculation method. After 2 days, cells were stained with cell viability dye, anti-CD4-APC and anti-p24-FITC, and measured by flow cytometry. At day 17, CD4$^+$ T cells were isolated using Dynabeads CD4 Positive Isolation Kit. At day 18, CRISPR-Cas9 knockout was generated as explained in section above. TracrRNA and guide RNA (IDT) were mixed and heated at 95 °C for 5 min. Cas9 (IDT) was added to the RNA mixture and incubated at room temperature for 20 min. CD4$^+$ T cells were washed with PBS and resuspended in Buffer T of Neon Transfection System 10 ml kit (Thermo Fisher). Pre-assembled Cas9-gRNA protein-RNA complexes were electroporated into cells using Neon equipment (Thermo Fisher). A CXCR4 gRNA was always

included as an additional control. After 2 days, live cells and CXCR4 staining were performed and western blot performed on ADAP1, p65/NF-κB, Ctrl cells to evaluate knockout efficiency. Cells were then seeded into 96-well plate, treated with or without 10 ng/ml PMA for 2 days. Luciferase of supernatant (Nano-Luc) was measured using Promega NanoLuciferase kit.

**RNA extraction and RT-qPCR assay**. Isolation of total RNA was done using the Quick-RNA miniprep kit (Zymo, R1055). RNA quality was assessed by computing RIN index (RNA Integrity Number) on 2200 Tapestation (Agilent, 5067-5576) RIN > 9.5 was standard cutoff. First strand cDNA synthesis was done by incubating 1 μg total RNA with 2.5 mM Oligo dT$_{(18)}$ and 0.5 mM dNTP mix (NEB, N0447L) for 5 min at 70 °C before placing on ice for 2 min. Next, 1X M-MuLV Reverse Transcriptase buffer (NEB, M02553L) and 10 U/ml of M-MuLV Reverse Transcriptase (NEB, M02553L) was added to each sample and incubated for 1 h at 42 °C. The reaction was inactivated for 10 min at 70 °C and samples were then diluted 1:5 with H$_2$O. For each qPCR sample well, 5 μl SYBR Green, 1 μl each primer (10 mM stock), 0.5 μl cDNA, and 3.5 μl H$_2$O were used in a 96-well plate. Samples were amplified 40 cycles using the Applied Biosystems 7500 Fast Real-Time PCR System. Ct values were obtained and fold change of the target mRNA levels relative to control was calculated as $2^{-\Delta\Delta Ct}$.

Primer sets used:

*ADAP1*: Fwd 5′-GAGCTGCTCGGGAATCCACC-3′, and Rev 5′-CTCGAAGG AGCTGGCAGTCG-3′

*RPL19*: Fwd 5′-ATCGATCGCCACATGTATCA-3′, and Rev 5′-GCGTGCTT CCTTGGTCTTAG-3′.

**Cell growth assay**. For cell growth assays, 50,000 $T_M$ were washed with 1X PBS and resuspended in RPMI-1640/10% FBS and seeded in triplicate using a multi-channel pipette into 96-well plates at day 0 in the absence of IL2. Cells were treated with either DMSO, 25 ng/ml PMA, 1 μM Ionomycin, 25 ng/ml PMA + 1 μM Ionomycin, or Dynabeads human T-activator anti-TCR/anti-CD28 (Gibco, 11132) (1 bead/cell) in a final volume of 0.1 ml. An additional 0.1 ml of media was added on day 4, for final volume of 0.2 ml to prevent overcrowding. On days 1, 2, 4, 6, and 9, cells were pelleted and thoroughly resuspended in 0.1–0.2 ml then counted using a hemocytometer to calculate cells/ml.

**Treatment of Jkt-HIVLuc cells with LRAs**. Lentiviruses were generated and cells transduced as described above. After 30 h, $2 \times 10^5$ transduced cells per sample were treated with increasing amounts (0–500 mM) of LRAs (JQ1 (APExBIO, A8181), TSA (Sigma, T8552), SAHA (APExbio, A4084)) in replicates. Cells were incubated in a final volume of 200 μl for 18 h at 37 °C. Firefly luciferase reporter activity was measured by incubating ~$2 \times 10^5$ cells in 25 μl lysis buffer (Promega, E1531) for 5 min while orbital shaking and using 20 μl of lysate with 20 μl of Luciferase Kit substrate (Promega, E1501).

**Reporting summary**. Further information on research design is available in the Nature Research Reporting Summary linked to this article.

## Data availability

The data that support this study are available from the corresponding author upon reasonable request. The scRNAseq raw and processed sequencing data used in this study are available in the NCBI Gene Expression Omnibus under accession number GSE169339. The proteomics data used in this study are available in the MassIVE under the ID MSV000088765 [https://massive.ucsd.edu/ProteoSAFe/dataset.jsp?task=0ce90f6187714960af4b828a695b7b35]. All other data generated in this study are provided in the main text, Supplementary Information, and Source Data File. Source data are provided with this paper.

## Code availability

The custom R script used for scRNAseq analysis has been deposited in Zenodo repository[52] and is publicly available at: https://github.com/bioinformatics-jeonlee/ADAP1_promotes_HIV-1_reactivation_scRNA-seq_analysis.

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

## Acknowledgements

We fully appreciate the UTSW McDermott Center Sequencing Core for assistance in scRNAseq experiments and A. Lemoff at the UTSW Proteomics Core for assistance with proteomics experiments. The authors appreciate the assistance of the UTSW Live Cell Imaging Facility, a shared resource of the Harold C. Simmons Comprehensive Cancer Center, supported in part by an NCI Cancer Center Support Grant, 1P30 CA142543-01. We thank J. Schoggins and lab for the vector pTRIP.CMV.IVSB.IRES.TagRFP. We thank M. Mettlen for guidance on the cell fractionation protocol. We also thank the rest of the D'Orso lab for valuable feedback throughout the project. This research was supported in part by US National Institutes of Health (NIH) under award numbers R01AI114362 and R21AI155071 (to I.D). N.-G.P.R. was supported by NIH Pharmacological Sciences Training grant GM007062 and a 2016 pre-doctoral fellowship from the Ford Foundation. J.L. was supported by the Cancer Prevention Research Institute of Texas (CPRIT; RP150596). V.P. was supported by NIH grant AI143567-02. K.D.W. was supported by NIH R01CA244341-01 and the Cancer Prevention Research Institute of Texas (CPRIT) RP170373.

## Author contributions

N.-G.P.R. and I.D. conceptualized the project, prepared the figures, and wrote the manuscript with suggestions from all co-authors. N.-G.P.R. generated Jkt-HIVLuc model, carried out screens, Jurkat transduction and luciferases assays, primary cell isolation and their respective assays, immunofluorescence, immunoprecipitations, RAS$^{GTP}$ ELISA, cell growth, flow cytometry, and qPCR experiments with their respective analysis. J.L. provided in depth scRNAseq analysis. Y.Z. isolated, infected, and carried out ex vivo model of latency experiments supervised by V.P. The Alpha and GTP hydrolysis assays and analysis were carried out by L.L. and B.D. supervised by K.D.W. The ORF library was provided by N.M.A. from which D.C. pooled cDNA and generated lentiviral pools for screen 1; N.-G.P.R. generated lentiviral pools for screen 2. A.C. purified recombinant proteins used in biochemical assays.

## Competing interests

The authors declare no competing interests.
