## [Peer Review File · Nature Communications]

ADAP1 promotes latent HIV-1 reactivation by selectively tuning KRAS–ERK–AP-1 T cell signaling-transcriptional axisReviewers' Comments:

Reviewer #1:

Remarks to the Author:

The paper from Ramirez et al. utilizes a gain of function genomic screen for reversal of HIV-1 latency to identify ADAP1 as factor involved in HIV-1 transcriptional induction. In addition to gain and loss of ADAP1 expression experiments to address potential function, they use a variety of approaches including primary latency models, Mass Spec, single cell RNA sequencing, microscopy and computational methods to suggest that ADAP1 coordinates T cell signaling upon stimulation. A strength of the paper are data demonstrating that ADAP-1 may "tune" KRAS and downstream signaling, including PKC θ and AP-1, is a strength. This role as an adapter or facilitator would also be consistent with some of the modest downstream signaling and transcriptional outcomes which are typically about two-fold. As might be expected, even slight changes in T cell signaling and cell proliferation will influence HIV expression. The data are in general supportive of the concept that ADAP1 is modestly influencing T cell signaling and HIV-1 transcription. Criticisms include over interpretation of data and lack of clarity concerning the expression pattern of ADAP1. Specific comments are below.

1. The authors conclude that the HIV is co-opting this ADAP-1-KRAS-AP-1 pathway. This is overstated since the virus does not appear to directly target ADAP1 or any downstream events. Rather, ADAP1 functions by creating an intracellular environment which is permissive for HIV-1 transcription.
2. Changes in HIV-1 transcription are quite modest, less than two fold induction when ADAP-1 levels are manipulated, especially considering that they observe greater than 50-fold increase with TNF α treatment.
3. If ADAP1 is acting through signaling, it would be interesting to see if downstream events can act in synergy with other latency reversing agents, such as HDACi or BET inhibitors, especially since the modest effects that are observed do not seem to suggest that ADAP1 would be a relevant direct target for latency reversal.
4. A minor point. Figure 3 and the description of this experiment, they should indicate that they knocked down the p65 subunit of NF- κ B.
5. The argument that knocking down ADAP1 with CRISPR leads to a 1.3X increase in basal HIV transcription and therefore plays a role in latency is speculative and this as well as well as modest baseline changes in AP-1 subunits should be part of the discussion since this is based on initial observations.
6. Figure 2 suggest that ADAP1 mRNA and protein are induced from modest baseline expression in memory cells to robust expression in activated cells. However, imaging in Figure 4 and western blots in figure 6 and extended data indicate that ADAP1 is expressed in memory cells, maybe even at comparable levels to activated cells. It is also difficult to appreciate if ADAP-1 has differential localization at the cell membrane raising questions about post-translational modifications. If present and at the membrane, then ADAP1 might be a target of kinases recruited to the signalosome. Insights into how ADAP1 is regulated in Tm and activated cells is critical to understanding how it is regulating the T cell signaling cascade.
7. The knockdown of ADAP1 and its impact on downstream events (Fig 6 and extended data) as determined by western blots for three donors seem variable and modest and are maybe less definitive as presented in the manuscript.
8. Minor point GWAS data seems tangential

Reviewer #2:

Remarks to the Author:

In this manuscript, Ramirez and collaborators identified ADAP1 as a novel modulator of HIV-1 proviral fate. In addition to controlling active versus latent infection, ADAP1 is critical at inducing transcriptional programs upon T cell stimulation, suggesting that HIV takes advantage of this activity to prevent latency. Through elegant high-throughput, genetic, biochemical, transcriptomic, and proteomic analyses, the authors effectively uncovered the mechanism by which ADAP1 plays such an important role. In particular, ADAP1 amplifies T cell signaling of the ERK-AP-1 axis by physically interacting and stimulating KRAS GTPase activity. In fact, ADAP1 reduction decreases gene expression after T cell stimulation, preventing latent HIV reactivation. Identification of ADAP1 not only fills an important knowledge gap in our understanding of the dependence of HIV on T cell signaling for active infection, but also identifies ADAP1 as a new target to reactivate the latent reservoir in shock and kill strategies. Overall, this is a very exciting discovery. I only have minor comments for the authors

1. The authors identified ADAP2 as another member of the ADAP family, which shares 55% identity to ADAP1. Since ADAP2 has no effect on latency reactivation, have the authors considered making chimeric mutants between ADAP1 and ADAP2 to map the domains in ADAP1 responsible for latency reactivation?

2. Figure 3, panel B. A western blot panel showing the loss of CXCR4 expression in cells knocked out of CXCR4 would be helpful

Point-by-point responses to the reviewer's comments

Reviewer 1

The paper from Ramirez et al. utilizes a gain of function genomic screen for reversal of HIV-1 latency to identify ADAP1 as factor involved in HIV-1 transcriptional induction. In addition to gain and loss of ADAP1 expression experiments to address potential function, they use a variety of approaches including primary latency models, Mass Spec, single cell RNA sequencing, microscopy and computational methods to suggest that ADAP1 coordinates T cell signaling upon stimulation. A strength of the paper are data demonstrating that ADAP-1 may “tune” KRAS and downstream signaling, including PKC θ and AP-1, is a strength. This role as an adapter or facilitator would also be consistent with some of the modest downstream signaling and transcriptional outcomes which are typically about two-fold. As might be expected, even slight changes in T cell signaling and cell proliferation will influence HIV expression. The data are in general supportive of the concept that ADAP1 is modestly influencing T cell signaling and HIV-1 transcription. Criticisms include over interpretation of data and lack of clarity concerning the expression pattern of ADAP1. Specific comments are below.

Response

We would like to thank the reviewer for the time spent reviewing our manuscript and for providing constructive criticisms to improve the presentation of our work.

Point 1

The authors conclude that the HIV is co-opting this ADAP1-KRAS-AP-1 pathway. This is overstated since the virus does not appear to directly target ADAP1 or any downstream events. Rather, ADAP1 functions by creating an intracellular environment which is permissive for HIV-1 transcription.

Response

We agree with the reviewer, and we have now revised the following statements:

“Our combined experimental approach defines ADAP1 as an unexpected tuner of T cell programs facilitating HIV-1 latency escape. (Abstract **pg. 2**)”

“Here, we present the discovery and detailed studies of ADAP1 as a previously unappreciated tuner of T cell signaling-transcriptional programs facilitating HIV-1 to escape latency.” (**pg. 4**).

Point 2

Changes in HIV-1 transcription are quite modest, less than two-fold induction when ADAP-1 levels are manipulated, especially considering that they observe greater than 50-fold increase with TNF- α treatment.

Response

We agree with the reviewer that changes in HIV-1 gene expression upon ADAP1 ectopic expression in the Jurkat model of latency (Jkt-HIVLuc) are modest. However, this is a caveat of the system and does not necessarily negate ADAP1's relevance. Supporting ADAP1's importance to T cell signaling and downstream gene expression, we would like to point out the following:

Point 2a. Modest ADAP1 effect on HIV-1 gene expression.

The gain-of-function approach in the Jurkat model of latency was done in the absence of T cell stimulation where a modest change upon single factor ectopic expression would not be surprising. Consistently, other relevant hits from the gain-of-function screen (e.g., CD80 and CD86 (**Figure 1e**)) behaved like ADAP1 in terms of potency increasing HIV-1 transcription, yet both are ligands of the CD28 co-receptor and well established to promote T cell signaling (PMID: 14647476) and latent HIV-1 reactivation (PMID: 8099629 and 24706775).

Additionally, we have performed an experiment whereby ectopic expression of p65/NF- κ B (arguably one of the most relevant factors known to promote HIV-1 transcription) in the Jurkat model of latency (Jkt-HIVLuc) does not have a positive effect on HIV-1 transcription (see **Figure below**), unlike ADAP1-mediated dose-dependent activation. As such, the apparent modest effect upon single factor ectopic expression relies on the factor's mechanism of action and does not necessarily discount its importance, as indicated above. As opposed to the gain-of-function approach, a complementary and more indicative measure of ADAP1's functional relevance would be the loss-of-function approach in the primary model of latency demonstrating that partial loss of ADAP1 in the T cell population dampened latent HIV-1 reactivation by ~40-50% in three donors (**Figure 3c**), near to a reduction in latent HIV-1 reactivation comparable to loss of p65/NF- κ B.

Figure (top): Fold luciferase activity of Jkt-HIVLuc cells transduced with increasing lentiviruses expressing GFP (negative control), ADAP1, or p65 subunit of NF- κ B. Data represents mean \pm s.d. of 3 independent experiments (n = 3) [two-way ANOVA test with multiple comparisons to respective dose GFP]. * P < 0.05, ** P < 0.01, *** P < 0.001, ns= not significant. **(bottom):** Western blot of Jkt-HIVLuc cells transduced with lentiviruses in top panel (showing highest dose) and probed with the indicated antibodies.

Point 2b. Comparison of ADAP1 vs TNF- α mechanisms of action and potencies.

It is noteworthy that ADAP1 and TNF- α have very different modes of action and therefore potency. TNF- α is a ligand that activates a signaling cascade and ADAP1 is one regulatory component of a signaling cascade (like p65 within the TNF- α signaling). Given this reason, comparing the latency reversal potential of TNF- α and ADAP1 ectopic expression can be misleading as it does not discount the relevance of ADAP1, and its mechanism of action as explained in **point 2a** and thoroughly addressed in the manuscript.

Point 3

If ADAP1 is acting through signaling, it would be interesting to see if downstream events can act in synergy with other latency reversing agents, such as HDACi or BET inhibitors, especially

since the modest effects that are observed do not seem to suggest that ADAP1 would be a relevant direct target for latency reversal.

Response

We thank the reviewer for the suggestion. We have now tested ADAP1's ability to act in synergy with commonly used latency reversing agents (LRAs such as SAHA, TSA, JQ1) in the Jurkat model of latency (Jkt-HIVLuc) and found ADAP1 does not work in synergy (see **Figure below**). Given the negative data we decided not to include it in the revised manuscript. Yet, to test whether ADAP1 is a relevant direct target for latency reversal approaches, as suggested by the reviewer, would require an ADAP1 specific drug yet to exist, which is well beyond the scope of this single study.

Further, as explained under **Point 2**, ADAP1's modest effect does not necessarily discount it from being a relevant target given ADAP1 works in synergy with KRAS to further induce HIV-1 transcription (**Figure 6e**). Because KRAS is difficult to target, and ADAP1 tunes KRAS, the possibility to indirectly interfere with KRAS by specifically targeting ADAP1 is worth investigating in future research. The fact that ADAP1 inactivates KRAS through GTP-to-GDP conversion in resting T cells may suggest ADAP1 could be targeted to promote latency reversal, also opening an interesting scenario for future research.

Figure: Fold luciferase activity of Jkt-HIVLuc cells transduced with lentiviruses expressing GFP (negative control) or ADAP1 and treated with increasing LRA concentrations: JQ1 (**top left**), SAHA (**top right**), and TSA (**bottom**).

Point 4

A minor point. Figure 3 and the description of this experiment, they should indicate that they knocked down the p65 subunit of NF- κ B.

Response

This is a good point. As suggested, we have now fixed the label in revised **Figure 3** and in the Results description (**pg. 7**).

Point 5

The argument that knocking down ADAP1 with CRISPR leads to a 1.3X increase in basal HIV transcription and therefore plays a role in latency is speculative and this as well as well as modest baseline changes in AP-1 subunits should be part of the discussion since this is based on initial observations.

Response

We thank the reviewer for the comment and agree that since these are initial observations, they are more appropriate for the Discussion. As such, we have also removed the following speculative statements from Results:

“This observation suggests ADAP1 may also participate in latency maintenance in the absence of stimulation.” (pg. 7).

“...further suggesting ADAP1 also participates in the negative regulation of AP-1 during the effector-to-memory transition.” (pg. 10)

Point 6

Point 6a. *ADAP1 expression levels.* Figure 2 suggest that ADAP1 mRNA and protein are induced from modest baseline expression in memory cells to robust expression in activated cells. However, imaging in **Figure 4** and western blots in figure 6 and extended data indicate that ADAP1 is expressed in memory cells, maybe even at comparable levels to activated cells.

Point 6b. *ADAP1 subcellular localization.* It is also difficult to appreciate if ADAP1 has differential localization at the cell membrane raising questions about post-translational modifications. If present at the membrane, then ADAP1 might be a target of kinases recruited to the signalosome. Insights into how ADAP1 is regulated in Tm and activated cells is critical to understanding how it is regulating the T cell signaling cascade.

Response

We thank the reviewer for these two points as they allow us to provide the needed clarifications to improve presentation of our data to the reader.

Point 6a. *Discrepancies in ADAP1 mRNA and protein expression levels.*

The reviewer noted confusion about ADAP1 expression levels not matching in Figures 2 and 4. Therefore, to make things clearer for the reader, we have now included new data of ADAP1 RNA and protein expression levels in a time course stimulation in two separate donors to clarify suggested discrepancies in ADAP1’s expression levels (new **Figures 2d-e**).

Specifically, while **Figures 2b-c** RT-qPCR and western blot data compare ADAP1 mRNA and protein levels at 0hr vs 24hr post-stimulation, microscopy images in **Figure 4a** were taken at 0hr and 4hr post-stimulation where the levels of ADAP1 protein are similar if not identical. This result is in line with similar levels of immunoprecipitated ADAP1 in unstimulated and 1hr stimulated primary CD4⁺ T cells (**Figure 4f**). Finally, levels of ADAP1 in memory T cells in **Figure 6f** and related extended data cannot be compared to activated levels since these are different membranes and exposure times vary. Finally, we have improved the description of these experiments in the main text by explicitly stating the time points analyzed. These new data will help guide the reader when comparing figures and experimental time points.

Point 6b. *ADAP1 subcellular localization.*

We agree with the reviewer that it is difficult to appreciate ADAP1 subcellular localization at the cell membrane given T cell morphology being largely nuclear and minimally cytoplasmic.

The main point to address here is to define how ADAP1 plasma membrane localization and/or post-translational modifications (PTMs) induced upon T cell stimulation could regulate its activating cell signaling function. ADAP1 could be: 1) located at the plasma membrane in both resting and activated T cells or 2) re-localized to the plasma membrane only in activated T cells. In the first model, ADAP1 is already at the membrane prior to stimulation and may be activated

through PTMs induced upon T cell stimulation. In the second model, ADAP1 relocalizes to the plasma membrane where it may or may not acquire PTMs upon T cell stimulation.

To start addressing these points we took two orthogonal approaches. First, to assess ADAP1's subcellular localization in memory cells before and after T cell stimulation, we now implemented an accepted cell fractionation protocol in the signaling field (PMID: 32657003). The new data reveal that in resting conditions, ADAP1 is in supernatant fractions (S) corresponding to cytoplasm (based on GAPDH localization); and under stimulation, ADAP1 is partially re-localized to pellet fractions (P) corresponding to plasma membrane (based on CD3 ϵ localization) (see new **Figure 4b** and data description in **pg. 8**). Note the result that ADAP1 levels are similar at resting and early time points (1 and 4 hr) post-stimulation, but sharply increase by 24 hr post-stimulation, which is consistent with ADAP1 expression pattern (see **Point 6a response**).

Second, to address whether ADAP1 acquires PTMs upon T cell stimulation, we have re-analyzed the IP-tandem mass spectrometry data to search for ADAP1's PTMs and found ADAP1 does not have PTMs at the time points (unstimulated and 1hr) tested. Notably, previous studies have revealed ADAP1 can be phosphorylated by PKC α at Ser87 and Thr276 *in vitro* (PMID: 12893243) with suggestion that these sites are targets of all PKC isoforms, thus opening the possibility PTMs induced upon T cell stimulation may contribute to ADAP1 activation. We generated alanine point mutants (S87A and T276A) to test whether these sites are necessary for ADAP1 mediated HIV-1 activity. From the **figure below**, neither point mutant dampened ADAP1's ability to induce HIV-1 gene expression in the Jkt-HIVLuc latency model. It is very well possible ADAP1 may be phosphorylated at different time points but identifying the sites, the kinase(s), and the mechanisms of regulation of ADAP1 activity are beyond the scope of this single study and will take years to properly address.

Collectively, our data support the model whereby ADAP1, which is mainly cytoplasmic in the resting T cell state, increases plasma membrane interaction/retention upon stimulation, thus fully addressing if ADAP1 has differential localization in the two T cell states. Given these new data, we have now revised the model in **Figure 8** to illustrate ADAP1 is localized to the plasma membrane upon stimulation where it forms part of the immune signalosome.

Figure legend: **(left)** Fold luciferase activity of Jkt-HIVLuc cells transduced with lentiviruses expressing GFP (mock), wild-type ADAP1, or ADAP1 S87A point mutant. **(right)** Fold luciferase activity of Jkt-HIVLuc cells transduced with lentiviruses expressing GFP (mock), wild-type ADAP1, or ADAP1 T276A point mutant.

Point 7

The knockdown of ADAP1 and its impact on downstream events (Fig 6 and extended data) as determined by western blots for three donors seem variable and modest and are maybe less definitive as presented in the manuscript.

Response

We appreciate the reviewer's point and take the opportunity to clarify the donor-to-donor variability as it relates: 1) to ADAP1 knockout efficiency and 2) its impact on downstream signaling (e.g., ERK1/2 phosphorylation) and transcription activation (e.g., *FOS* induction) events.

First, regarding ADAP1's knockdown efficiency, experiments using primary cells are subject to donor-to-donor variabilities. One must consider the expected differences in CRISPR-Cas9 efficiency for ADAP1 depletion (between 50-80% knockdown) across donors which is far from being complete due to expected technical reasons.

Second, to mitigate the expressed concern on the impact on downstream signaling and transcription activation, we now analyzed a fourth donor and provided an expanded data analysis for the combined donors (new **Extended Data Figure 6**). Specifically, the collective data analysis in the four donors indicates that, despite donor-to-donor variability, partial ADAP1 knockdown in the primary CD4+ T cell population partially dampens activation of the KRAS–ERK1/2–AP-1 pathway: overall decreased phosphorylated ERK1/2 by 30 min in four donors: donor 1 (~4.4-fold), donor 2 (~1.5-fold), donor 3 (~1.3-fold), and donor 4 (~3.8-fold) (**Extended Data Figure 6e-h**), and overall decreased Fos induction at 120 min in four donors: donor 1 (~2.5-fold), donor 2 (~2.9-fold), donor 3 (~3.2-fold), and donor 4 (>10-fold) (**Extended Data Figure 6i-j**). Although donors 1-3 (male) were comparable in age (40-73 years old), donor 4 was a 16-year-old boy, perhaps explaining the larger response and Fos induction. These details (now included in the figure legend) exemplify an added complexity that may trigger donor-to-donor variability. Nonetheless, this new analysis reveals that data from four donors are collectively consistent and statistically significant, and agree with other data in the parental manuscript strongly indicating ADAP1 knockdown has indeed an impact on downstream T cell events.

Given the above reasons, we propose that the observed decreases in phosphorylated ERK1/2 (ranging from ~1.3 to ~4.4-fold depending on donor) and Fos levels (ranging from ~2.5 to greater than 10-fold depending on donor) are not modest and represent the biologic scenario of ADAP1 partial regulation of the RAS–ERK1/2–AP-1 cell signaling cascade. Consistent with our proposal, RAS-ERK1/2 are targets of multiple TCR/CD28 stimulated pathways including LAT-PLC-RASGRP, LAT-Grb2-SOS (reviewed in PMID: 24027568), and PLC-PKC-RasGRP (PMID: 15899849). As such, it is reasonable to expect levels of active (phosphorylated) ERK1/2 and Fos include ADAP1-dependent and -independent effects. This idea is supported by the dampening of KRAS–ERK1/2–AP-1 signaling rather than complete ablation, consistent with our model that ADAP1 is a “tuner” of the system.

Point 8

Minor point GWAS data seems tangential.

Response

We agree with the reviewer's comment, and we have decided to eliminate the GWAS analysis. (pg. 13-14).

Reviewer 2

In this manuscript, Ramirez and collaborators identified ADAP1 as a novel modulator of HIV-1 proviral fate. In addition to controlling active versus latent infection, ADAP1 is critical at inducing transcriptional programs upon T cell stimulation, suggesting that HIV takes advantage of this activity to prevent latency. Through elegant high-throughput, genetic, biochemical, transcriptomic, and proteomic analyses, the authors effectively uncovered the mechanism by which ADAP1 plays such an important role. In particular, ADAP1 amplifies T cell signaling of the ERK-AP-1 axis by physically interacting and stimulating KRAS GTPase activity. In fact, ADAP1 reduction decreases gene expression after T cell stimulation, preventing latent HIV reactivation. Identification of ADAP1 not only fills an important knowledge gap in our understanding of the dependence of HIV on T cell signaling for active infection, but also identifies ADAP1 as a new target to reactivate the latent reservoir in shock and kill strategies. Overall, this is a very exciting discovery. I only have minor comments for the authors.

Response

We would like to thank the reviewer for the time spent reviewing our manuscript and for providing constructive criticisms to improve the presentation of our work.

Point 1

The authors identified ADAP2 as another member of the ADAP family, which shares 55% identity to ADAP1. Since ADAP2 has no effect on latency reactivation, have the authors considered making chimeric mutants between ADAP1 and ADAP2 to map the domains in ADAP1 responsible for latency reactivation?

Response

This is a good point. As suggested by the reviewer, we have now created the following chimera (ADAP2_{GAP}-ADAP1_{PH1PH2}) to attempt mapping domains in ADAP1 responsible for latent HIV-1 reactivation. However, the chimera expressed poorly as did a truncated ADAP1 (ADAP1_{PH1PH2}) (see **Figure** below). This result can be attributed to the need of intramolecular interactions among residues in the GAP and PH1 domains required for protein stability, consistent with structural data (PMID: 21057110). Given these data, and the functional importance of GAP and PH1 domains for ADAP1's GAP activity (PMID: 21057110) we contend ADAP1-ADAP2 domain swapping experiments may compromise ADAP1 structure and inter-domain contacts required for its catalytic function. Addressing the specific details will require structure-function analysis that although interesting are beyond the scope of this work.

Figure legend: (left) Western blot of transfected HEK293T with empty vector, wild-type ADAP1, or the chimera ADAP2_{GAP}-ADAP1_{PH1PH2} and probed with the indicated antibodies. (right) Western blot of transfected HEK293T with GFP, full-length wild-type ADAP1, or truncated ADAP1_{PH1PH2} and probed with the indicated antibodies.

Point 2

Figure 3, panel B. A western blot panel showing the loss of CXCR4 expression in cells knocked out of CXCR4 would be helpful.

Response

This is a good point and we have now incorporated the requested control in **Figure 3**.

Reviewers' Comments:

Reviewer #1:

Remarks to the Author:

The authors thoughtfully responded to the concerns by adding data and more clearly discussing their results. One very minor point, I think the LRA data, even though negative, is interesting and should be considered at least as a discussion point.

Reviewer #2:

Remarks to the Author:

The authors have satisfactorily addressed my questions. This new version of the manuscript is significantly improved. Hence, I am in favor of accepting the article for publication

Point-by-point responses to the reviewer's comments

Reviewer 1

The authors thoughtfully responded to the concerns by adding data and more clearly discussing their results. One very minor point, I think the LRA data, even though negative, is interesting and should be considered at least as a discussion point.

Response

We would like to thank the reviewer for the time spent re-reviewing our manuscript and suggesting its acceptance. We agree with the minor point. As such, we have now added the data to new Supplementary Fig. 8 and discussed the data in **pg. 18**.

Reviewer 2

The authors have satisfactorily addressed my questions. This new version of the manuscript is significantly improved. Hence, I am in favor of accepting the article for publication.

Response

We would like to thank the reviewer for the time spent re-reviewing our manuscript and suggesting its acceptance.